# ConR: Contrastive Regularizer for Deep Imbalanced Regression

**Mahsa Keramati**[1,2,*] **Lili Meng**[1]**, R. David Evans**[1]
[1] Borealis AI, [2] School of Computing Science, Simon Fraser University
`mkeramat@sfu.ca, lili.meng@gmail.com, dave.evans@borealisai.com`

## Abstract

Imbalanced distributions are ubiquitous in real-world data. They create constraints on Deep Neural Networks to represent the minority labels and avoid bias towards majority labels. The extensive body of imbalanced approaches address categorical label spaces but fail to effectively extend to regression problems where the label space is continuous. Local and global correlations among continuous labels provide valuable insights towards effectively modelling relationships in feature space. In this work, we propose **ConR**, a contrastive regularizer that models global and local label similarities in feature space and prevents the features of minority samples from being collapsed into their majority neighbours. **ConR** discerns the disagreements between the label space and feature space, and imposes a penalty on these disagreements. **ConR** addresses the continuous nature of label space with two main strategies in a contrastive manner: incorrect proximities are penalized proportionate to the label similarities and the correct ones are encouraged to model local similarities. **ConR** consolidates essential considerations into a generic, easy-to-integrate, and efficient method that effectively addresses deep imbalanced regression. Moreover, **ConR** is orthogonal to existing approaches and smoothly extends to uni- and multi-dimensional label spaces. Our comprehensive experiments show that **ConR** significantly boosts the performance of all the state-of-the-art methods on four large-scale deep imbalanced regression benchmarks. Our code is publicly available in https://github.com/BorealisAI/ConR.

## 1 Introduction

Imbalanced data distributions, which are common in real-world contexts, introduce substantial challenges in generalizing conventional models due to variance across minority labels and bias to the majority ones (Wang et al., 2021b; Gong et al., 2022; Buda et al., 2018). Although there are numerous works on learning from imbalanced data (Chawla et al., 2002; Cui et al., 2021; Jiang et al., 2021), these studies mainly focus on categorical labels. Continuous labels are potentially infinite, high-dimensional and hard to bin semantically (Ren et al., 2022). These characteristics impede the performance of imbalanced classification approaches on Deep Imbalanced Regression (DIR) (Yang et al., 2021).

Continuous labels result in underlying local and global correlations, which yields valuable perspectives towards effective representation learning of imbalanced data (Gong et al., 2022; Shin et al., 2022). For instance, regression training on imbalanced data fails to model appropriate relationships for minority labels in the feature space (Yang et al., 2021). Yang et al. (2021) established an empirical example of this phenomenon on the age estimation task where the learned features of under-represented samples collapse to majority features. Several approaches tackle this issue by encouraging local dependencies (Yang et al., 2021; Steininger et al., 2021). However, these methods fail to exploit global relationships and are biased toward only learning representative features for majority samples, especially when minority examples do not have majority neighbours. RankSim (Gong et al., 2022) leverages global and local dependencies by exploiting label similarity orders in feature space. Yet, RankSim does not generalize to all regression tasks, as not all continuous label spaces convey order relationships. For example, for depth-map estimation from scene pictures, the complicated

---

[*]Work done during an internship at Borealis AI.

relationships in the high-dimensional label space are not trivially convertible to a linearly ordered feature space. **Given the importance of the correspondence between the label space and feature space for imbalanced regression, can we effectively transfer inter-label relationships, regardless of their complexity, to the feature space?**

We propose a method to enforce this correspondence: **ConR** is a novel **Con**trastive **R**egularizer which is based on infoNCE loss (Oord et al., 2018) but adapted for multi-positive pairs. While similar extensions are performed for classification tasks (Khosla et al., 2020b), **ConR** addresses continuous label space and penalizes minority sample features from collapsing into the majority ones. Contrastive approaches for imbalanced classification are mainly based on predefined anchors, decision boundary-based negative pair selection and imposing discriminative feature space (Cui et al., 2021; Wang et al., 2021a; Li et al., 2022). These are not feasibly extendible to continuous label spaces. **ConR** introduces continuity to contrastive learning for regression tasks. Fig 1 illustrates an intuitive example for **ConR** from the task of age estimation. There are images of individuals of varying ages, including 1, 21, 25, and 80 years. Age 1 and 80 are the minority examples, reflecting the limited

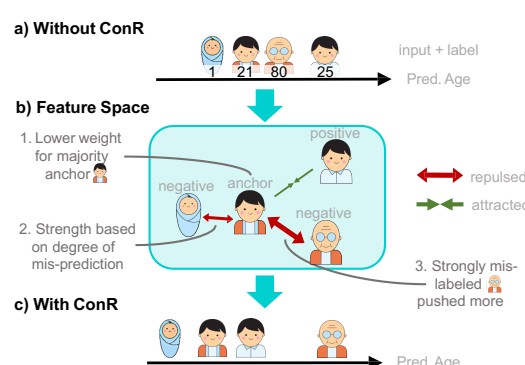

Figure 1: Key insights of **ConR**. a) Without **ConR**, it is common to have minority examples mixed with majority examples. b) **ConR** selects the sample with confusion around it as an anchor and adjusts the feature space with relative contrastive learning. c) Reduced prediction error.

number of images available for these age groups within the datasets. While 21 is a majority example, given the abundance of images around this age within the datasets. Without using **ConR**, similarities in the feature space are not aligned with the relationships in the label space. Thus, the minority samples' features collapse to the majority sample, leading to inaccurate predictions for minority ones that mimic the majority sample (Fig 1a). **ConR** regularizes the feature space by simultaneously encouraging locality via pulling together positive pairs and preserving global correlations by pushing negative pairs. The 21-year-old sample coexists within a region in the feature space alongside 1-year-old and 80-year-old samples. Thus, **ConR** 1) considers the 21-year-old sample as an anchor, and, 2) pushes negative pairs for minority anchors harder than majority examples to provide better separation in feature space. Furthermore, **ConR** 3) increases pushing power based on how heavily mislabelled an example is (Fig 1b.1 to Fig 1b.3). We demonstrate that **ConR** effectively translates label relationships to the feature space, and boosts the regression performance on minority samples (Fig 1c). Refer to A.1 for the empirical analysis on the motivation of **ConR**.

**ConR** implicitly models the local and global correlations in feature space by three main contributions to the contrastive objective: **1) Dynamic anchor selection:** Throughout the training, considering the learned proximity correspondences, **ConR** selects samples with the most collapses on the feature manifold as anchors. **2) Negative Pairs selection:** Negative pairs are sampled to quantify the deviation they introduce to the correspondence of similarities in feature and label space, thereby compensating for under-represented samples. **3) Relative pushing:** Negative pairs are pushed away proportional to their label similarity and the density of the anchor's label.

**ConR** is orthogonal to other imbalanced learning techniques and performs seamlessly on high-dimensional label spaces. Our comprehensive experiments on large-scale DIR benchmarks for facial age, depth and gaze estimation show that **ConR** strikes a balance between efficiency and performance, especially on depth estimation, which has a complicated and high-dimensional label space.

## 2 RELATED WORK

**Imbalanced classification.** The main focus of existing imbalanced approaches is on the classification task, while imbalanced regression is under-explored. The imbalanced classification methods are either data-driven or model-driven. Resampling (Chawla et al., 2002; Chu et al., 2020; Byrd & Lipton, 2019; Han et al., 2005; Jiang et al., 2021) is a data-driven technique that balances the input

data by either over-sampling (Chawla et al., 2002; Byrd & Lipton, 2019) the minority classes or under-sampling the majority ones (Han et al., 2005). Model-Aware K-center(MAK) (Jiang et al., 2021) over-samples tail classes using an external sampling pool. Another branch of data-driven approaches is augmentation-based methods. Mixup-based approaches linearly interpolate data either in input space (Zhang et al., 2017) or feature space (Verma et al., 2019) for vicinal risk minimization. RISDA (Chen et al., 2022) shares statistics between classes considering a confusion-based knowledge graph and implicitly augments minority classes. Model-driven methods such as focal loss (Lin et al., 2017) and logit-adjustment (Tian et al., 2020) are cost-sensitive approaches that regulate the loss functions regarding the class labels. To encourage learning an unbiased feature space, several training strategies including two-stage training (Kang et al., 2020), and transfer learning (Yin et al., 2019) are employed. As discussed further, though there has been much success in this area, there are many issues in converting imbalanced classification approaches to regression.

**Imbalanced regression.** Unlike classification tasks with the objective of learning discriminative representation, effective imbalanced learning in continuous label space is in lieu of modelling the label relationships in the feature space (Yang et al., 2021; Gong et al., 2022). Therefore, imbalanced classification approaches do not feasibly extend to the continuous label space. DenseLoss (Steininger et al., 2021) and LDS (Yang et al., 2021) encourage local similarities by applying kernel smoothing in label space. Feature distribution smoothing (FDS) (Yang et al., 2021) extends the idea of kernel smoothing to the feature space. Ranksim (Gong et al., 2022) proposes to exploit both local and global dependencies by encouraging a correspondence of similarity order between labels and features. Balanced MSE (Ren et al., 2022) prevents Mean Squared Error (MSE) from carrying imbalance to the prediction phase by restoring a balanced prediction distribution. The empirical observations in (Yang et al., 2021) and highlighted in VIR (Wang & Wang, 2023), demonstrate that using empirical label distribution does not accurately reflect the real label density in regression tasks, unlike classification tasks. Thus, traditional re-weighting techniques in regression tasks face limitations. Consequently, LDS (Yang et al., 2021) and the concurrent work, VIR propose to estimate effective label distribution. Despite their valuable success, two main issues arise, specifically for complicated label spaces. These approaches rely on binning the label space to share local statistics. However, while effective in capturing local label relationships, they disregard global correspondences. Furthermore, in complex and high-dimensional label spaces, achieving effective binning and statistics sharing demands in-depth domain knowledge. Even with this knowledge, the task becomes intricate and may not easily extend to different label spaces.

**Contrastive learning.** Contrastive Learning approaches are pairwise representation learning techniques that push away semantically divergent samples and pull together similar ones (He et al., 2020; Chen et al., 2020a; Khosla et al., 2020a; Kang et al., 2021a). Momentum Contrast (Moco) (He et al., 2020) is an unsupervised contrastive learning approach that provides a large set of negative samples via introducing a dynamic queue and a moving-averaged encoder; while SupCon (Khosla et al., 2020b) incorporates label information to the contrastive learning. Kang et al. (2021a) argue that in case of imbalanced data, SupCon is subject to learning a feature space biased to majority samples and proposed k-positive contrastive loss (KCL) to choose the same number of positive samples for each instance to alleviate this bias. Contrastive long-tailed classification methods train parametric learnable class centres (Cui et al., 2021; Wang et al., 2021a) or encourage learning a regular simplex (Li et al., 2022). However, these approaches cannot be used for regression tasks where handling label-wise prototypes is potentially complex. Contrastive Regression (CR) (Wang et al., 2022) adds a contrastive loss to Mean Squared Error (MAE) to improve domain adaptation for the gaze estimation task. CR assumes there is a correspondence between similarities in label space and the feature space. Regardless, when learning from the imbalanced data, the correspondence can't be assumed. Instead of the assumption of correspondence, **ConR** translates the relative similarities from the label space to the feature space. Barbano et al. (2023) proposed a relative pushing for the task of brain age prediction using MRI scans. This relative pushing doesn't consider imbalanced distribution, and the weights are learnable kernels that can introduce complexities to complex label spaces. Rank-N-Contrast (Zha et al., 2023) ranks samples and contrasts them against each other regarding their relative rankings to impose continuous representations for regression tasks. Like RankSim Gong et al. (2022), Rank-N-Contrast is designed based on the order assumption and cannot be used in all regression tasks.

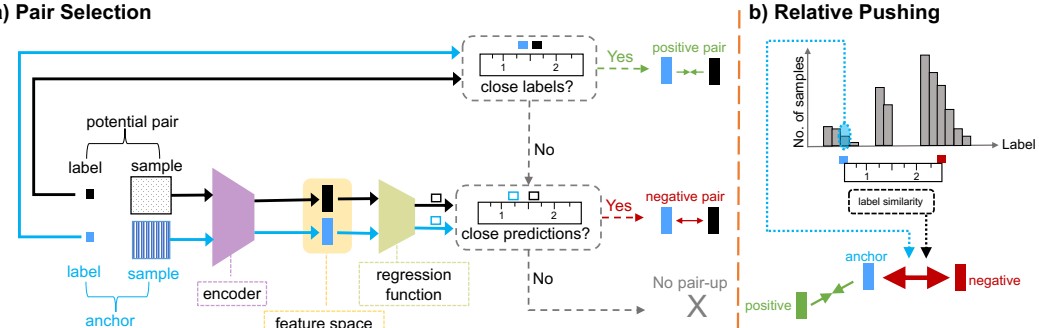

Figure 2: The framework of **ConR** is to translate label similarities to the feature space. a) Per each augmented sample, **ConR** selects positive and negative pairs with regard to the label similarities and prediction similarities. b) **ConR** pulls positive pairs together while pushing away negative pairs regarding their label similarities and label distribution for the anchor. In this way, the minority anchor pushes negative samples harder. The pushing weight is inversely relative to the label similarities.

## 3 METHOD

### 3.1 PROBLEM DEFINITION

Consider a training dataset consisting $N$ examples, which we denote as $\{(x_i, y_i)\}_{i=0}^{N}$, where $x_i \in \mathbb{R}^d$ is an example input, and $y_i \in \mathbb{R}^{d'}$ is its corresponding label. $d$ and $d'$ are the dimensions of the input and label, respectively. We additionally enforce that the distribution of the labels $\mathcal{D}_y$ deviates significantly from a uniform distribution. Given a model that consists of a feature encoder $\mathcal{E}(\cdot)$, and a regression function $\mathcal{R}(\cdot)$, the objective is to train a regression model $\mathcal{R}(\mathcal{E}(\cdot))$, such that the model output $\hat{y}_i = \mathcal{R}(\mathcal{E}(x_i))$ is similar to the true label $y_i$.

### 3.2 IMBALANCED CONTRASTIVE REGRESSION

In regression tasks, inter-label relationships unveil meaningful associations in feature space. However, learning from imbalanced data harms this correspondence since the features learned for minority samples share statistics with the majority samples, despite dissimilar labels (Yang et al., 2021). By incorporating label and prediction relationships into contrastive learning, **ConR** enforces appropriate feature-level similarities for modelling a balanced feature space.

**ConR** is a continuous variation of infoNCE. Initially, for creating diversity in examples, we perform problem-specific augmentations on each input $x_i$ to produce two augmented samples. We define the set of augmented examples from the input to be $\{(x_j^a, y_j)\}_{j=0}^{2N}$, where $x_j^a$ is an augmented input.

As illustrated in Fig. 2, **ConR** is a collaboration of pair selection and relative pushing. For each augmented sample, ConR first selects pairs. Next, in the case of at least one negative pairing, the sample is considered an anchor and contributes to the regularizing process of **ConR** by pulling together positive pairs and relatively repelling negative pairs.

#### 3.2.1 PAIR SELECTION

Given a pair of examples $(x_i^a, y_i)$ and $(x_j^a, y_j)$ from the augmented inputs (*labels* and *samples*, Fig. 2-a), each example is passed to the feature encoder $\mathcal{E}(\cdot)$ to produce feature vectors $z_i$ and $z_j$, and then to the regression function $\mathcal{R}(\cdot)$ for predictions $\hat{y}_i$ and $\hat{y}_j$ (*sample* through *regression function*, Fig. 2-a). The values of the predicted and actual labels are used to determine if the examples should be a positive pair, a negative pair, or unpaired (righthand side, Fig. 2-a).

To measure similarity between labels (or predictions) of augmented examples, **ConR** defines a similarity threshold $\omega$. Given a similarity function $Sim(\cdot, \cdot) \in \mathbb{R}$ (e.g., inverse square error), we define two labels (or two predictions) $y_i$ and $y_j$ as similar if $Sim(y_i, y_j) \geq \omega$. We denote this as $y_i \simeq y_j$. Iff two examples have similar labels $y_i \simeq y_j$, then they are treated as a positive pair. The

examples have dissimilar labels, but similar predictions $\hat{y}_i \simeq \hat{y}_j$, then they are treated as a negative pair. Otherwise, examples are unpaired.

**Anchor selection.** For each example $j$, $(x_j^a, y_j)$ We denote the sets of positive and negative pairs for this example, and their feature vectors as $K_j^+ = \{(z_p)\}_p^{N_j^+}$ and $K_j^- = \{(z_q)\}_q^{N_j^-}$, respectively, where $N_j^+$ is the number of positive examples and $N_j^-$ is the number of negative samples for example $j$. If $N_j^- > 0$, $(x_j^a, y_j)$ is selected as an anchor and contributes to the regularization process of **ConR**.

### 3.2.2 CONTRASTIVE REGULARIZER

For each example $j$, **ConR** introduces a loss function $\mathcal{L}_{ConRj}$. If example $j$ is not selected as an anchor, $\mathcal{L}_{ConRj} = 0$. Otherwise, $\mathcal{L}_{ConRj}$ pulls together positive pairs while pushing away negative pairs proportional to the similarity of their labels. As shown in Fig. 2-b, **ConR** pushes away negative samples with less label similarity to $y_j$ harder than negative samples with labels closer to the anchor $y_j$:

$$\mathcal{L}_{ConRj} = -\log \frac{1}{N_j^+} \sum_{z_i \in K_j^+} \frac{\exp(z_j \cdot z_i / \tau)}{\sum_{z_p \in K_j^+} \exp(z_j \cdot z_p / \tau) + \sum_{z_q \in K_j^-} \mathcal{S}_{j,q} \exp(z_j \cdot z_q / \tau)} \quad (1)$$

where $\tau$ is a temperature hyperparameter and $\mathcal{S}_{j,n}$ is a pushing weight for each negative pair:

$$\mathcal{S}_{j,q} = f_{\mathcal{S}}(\eta_j, Sim(y_j, y_q)), \quad (2)$$

and $y_q$ is the label of $x_q$ where $z_q = \mathcal{E}(x_q)$. $\eta_j$ is a pushing power for each sample $(x_j^a, y_j)$ that depends on label distribution $\mathcal{D}_y$ to boost the pushing power for minority samples (Fig. 2-b): $\eta_j \propto w_j$, where $w_j$ is a density-based weight for input $j$ derived from the empirical label distribution (e.g., inverse frequency). The function $f_{\mathcal{S}}$ computes the $\mathcal{S}_{j,q}$ to be proportionate to $\eta_j$ and inversely related to $Sim(y_j, y_q)$. Please refer to Appendix A.4 for the definition of $f_{\mathcal{S}}$.

Finally, $\mathcal{L}_{ConR}$ is the **ConR**'s regularizer value for the augmented example set:

$$\mathcal{L}_{ConR} = \frac{1}{2N} \sum_{j=0}^{2N} \mathcal{L}_{ConRj}. \quad (3)$$

To prevent the representations of minority labels from collapsing to the majority ones in deep imbalanced regression, $L_{sum}$ is optimized. $L_{sum}$ is weighed sum of $\mathcal{L}_{ConR}$ and a regression loss $\mathcal{L}_{\mathcal{R}}$ (e.g., mean absolute error) as below:

$$L_{sum} = \alpha \mathcal{L}_{\mathcal{R}} + \beta \mathcal{L}_{ConR} \quad (4)$$

### 3.3 THEORETICAL INSIGHTS

We theoretically justify the effectiveness of **ConR** in Appendix B. We derive the upper bound $\mathcal{L}_{ConR} + \epsilon$ on the probability of incorrect labelling of minority samples to be:

$$\frac{1}{4N^2} \sum_{j=0, x_j \in A}^{2N} \sum_{q=0}^{K_j^-} \log \mathcal{S}_{j,q} p(\hat{Y}_j | x_q) \leq \mathcal{L}_{ConR} + \epsilon, \quad \epsilon \overset{N \to \infty}{\to} 0 \quad (5)$$

where $A$ is the set of anchors and $x_q$ is a negative sample. $p(\hat{Y}_j | x_q)$ is the likelihood of sample $x_q$ with an incorrect prediction $y_q \in \hat{Y}_j = (\hat{y}_j - \omega, \hat{y}_j + \omega)$. $\hat{y}_j$ a prediction that is mistakenly similar to its negative pair $\hat{y}_q$. We refer to $p(\hat{Y}_j | x_q)$ as the probability of collapse for $x_q$. The Left-Hand Side (LHS) contains the probability of all collapses for all negative samples, which is bounded above by $\mathcal{L}_{ConR}$. Minimizing the loss, consequently minimizes the LHS, which causes either 1) the number of anchors decreases or 2) the degree of collapses is reduced. Each $p(\hat{Y}_j | x_q)$ is weighted by $\mathcal{S}_{j,q}$, which penalizes incorrect predictions proportional to severity. Thus, optimizing ConR decreases the probability of mislabelling proportional to the degree, and emphasizes minority samples.

## 4 EXPERIMENTS

### 4.1 MAIN RESULTS

**Datasets and baselines.**  We use three datasets curated by Yang et al. (2021) for the deep imbalanced regression problem: AgeDB-DIR is a facial age estimation benchmark, created based on AgeDB (Moschoglou et al., 2017). IMDB-WIKI-DIR is an age estimation dataset originated from IMDB-WIKI (Rothe et al., 2018). NYUD2-DIR is created based on NYU Depth Dataset V2 (Silberman et al., 2012) to predict the depth maps from RGB indoor scenes. Moreover, we create MPIIGaze-DIR based on MPIIGaze, which is an appearance-based gaze estimation benchmark. Refer to Appendix A.2 and A.4 for more dataset and implementation details. Please refer to Appendix A.3 for baseline details.

**Evaluation process and metrics.**  Following the standard procedure for imbalanced learning (Yang et al., 2021; Gong et al., 2022), we report the results for four shots: *All*, *few*, *median* and *many*. The whole test data is denoted by *All*. Based on the number of samples in the training dataset, *few*, *median* and *many* that have less than 20, between 20 and 100, and more than 100 training samples per label, respectively.

For AgeDB-DIR and IMDB-WIKI-DIR, the metrics are Mean-Absolute-Error (MAE), and Geometric Mean (GM). For NYUD2-DIR we use Root Mean Squared Error (RMSE) and Threshold accuracy ($\delta_1$) as metrics as in (Yang et al., 2021; Ren et al., 2022). Threshold accuracy $\delta_i$ is the percentage of $d_i$ that satisfies $max(\frac{d_1}{g_1}, \frac{g_1}{d_1}) < 1.25$, where for each pixel, $g_1$ is the ground truth depth value and $d_1$ is the predicted depth value. For MPIIGaze-DIR, we use Mean Angle Error (degrees). To calculate relative improvement of **ConR**, we compare the performance of each combination of methods including **ConR**, against the same combination without **ConR**. Each "Ours *vs. ...*" entry shows the average of these improvements (e.g. "Ours *vs.* LDS" is the average of the improvements of adding **ConR** to each combination of baselines that has LDS).

**Main results for age estimation.**  Table 1 and Table 2 show the results on AgeDB-DIR and IMDB-WIKI-DIR benchmarks, respectively. We compare the performance of DIR methods: FDS, LDS and RankSim with their regularized version by **ConR**. All results are the averages of 5 random runs. We observe that the performances of all the baselines are considerably boosted when they are regularized with **ConR**. In addition, **ConR** results in the best performance across both metrics and all shots for the AgeDB-DIR benchmark with leading MAE results of 6.81 and 9.21 on the overall test set and *few*-shot region, respectively. For IMDB-WIKI-DIR, **ConR** achieves the highest performance on 7 out of 8 shot/metric combinations with the best MAE results of 7.29 and 21.32 on the overall test set and *few*-shot region, respectively.

**Main results for depth estimation.**  To assess the effectiveness of **ConR** in more challenging settings where the label space is high-dimensional with non-linear inter-label relationships, we compare its performance with baselines: LDS, FDS and Balanced MSE on NYUD2-DIR depth estimation benchmark. For this task, the label similarity is measured by the difference between the average depth value of two samples. As shown in table 3, **ConR** alleviates the bias of the baseline toward the majority samples. **ConR** significantly outperforms LDS, FDS and Balanced MSE across all shots and both metrics with leading RMSE results of 1.265 on the overall test set and 1.667 on the *few*-shot region. Notably, RankSim cannot be used on the depth estimation task; however, **ConR** can smoothly be extended to high-dimensional label space with high performance. The reason is that order relationships are not semantically meaningful for all regression tasks with high-dimensional label space, such as depth estimation.

**Main results for gaze estimation.**  Table 4 compares the performance of **ConR** with three baseline methods: LDS, FDS and Balanced MSE on the MPIIGaze-DIR benchmark. As shown in table 4, **ConR** consistently improve the performance of all baselines for all shots and achieves the best Mean angular error of 5.63 (degrees) on the overall test set and 5.21 (degrees) on the *few*-shot region.

**Error reduction.**  In Fig. 3, we show the comparison between three strong baselines (LDS, FDS and Balanced MSE) and by adding **ConR** for NYUD2-DIR benchmark. It shows a consistent and notable error reduction effect by adding our **ConR** to the deep imbalanced learning. For more results and analysis on other datasets, please refer to Appendix A.5.

Table 1: **Main results for AgeDB-DIR benchmark**. Results are reported for the whole test data (*all*) and three other shots: *many*, *median*, and *few*. Each section compares a baseline with its regularized version of **ConR**. At the bottom of the table, the average improvements with respect to corresponding baselines without **ConR** are reported in Green. The best result of either a baseline or its regularized version by **ConR** is in **bold** (in column) and Red (across column). Baselines: (Gong et al., 2022).

| Metrics | MAE↓ | | | | GM↓ | | | |
|---|---|---|---|---|---|---|---|---|
| Methods/Shots | All | Many | Median | Few | All | Many | Median | Few |
| ConR-only (Ours) | 7.20 | 6.50 | 8.04 | 9.73 | 4.59 | 3.94 | 4.83 | 6.39 |
| LDS | 7.42 | 6.83 | 8.21 | 10.79 | 4.85 | 4.39 | 5.80 | 7.03 |
| LDS + **ConR** (Ours) | **7.16** | **6.61** | **7.97** | **9.62** | **4.51** | **4.21** | **4.92** | **5.87** |
| FDS | 7.55 | 6.99 | 8.40 | 10.48 | 4.82 | 4.49 | 5.47 | 6.58 |
| FDS + **ConR** (Ours) | **7.08** | **6.46** | **7.89** | **9.80** | **4.31** | **4.01** | **5.25** | **6.92** |
| RankSim | 6.91 | 6.34 | 7.79 | 9.89 | 4.28 | 3.92 | 4.88 | 6.89 |
| RankSim + **ConR** (Ours) | **6.84** | **6.31** | **7.65** | **9.72** | **4.21** | **3.95** | **4.75** | **6.28** |
| LDS + FDS | 7.37 | **6.82** | 8.25 | 10.16 | 4.72 | **4.33** | 5.50 | 6.98 |
| LDS + FDS + **ConR** (Ours) | **7.21** | 6.88 | **7.63** | **9.59** | **4.63** | 4.45 | **5.18** | **5.91** |
| LDS + RankSim | 6.99 | 6.38 | 7.88 | 10.23 | **4.40** | 3.97 | 5.30 | 6.90 |
| LDS + RankSim + **ConR** (Ours) | **6.88** | **6.35** | **7.69** | **9.99** | 4.43 | **3.87** | **4.70** | **6.51** |
| FDS + RankSim | 7.02 | 6.49 | 7.84 | 9.68 | 4.53 | 4.13 | 5.37 | 6.89 |
| FDS + RankSim+ **ConR** (Ours) | **6.97** | **6.33** | **7.71** | **9.43** | **4.19** | **3.92** | **4.67** | **6.14** |
| LDS + FDS + RankSim | 7.03 | 6.54 | 7.68 | 9.92 | 4.45 | 4.07 | 5.23 | 6.35 |
| LDS + FDS + RankSim + **ConR** (Ours) | **6.81** | **6.32** | **7.45** | **9.21** | **4.39** | **3.81** | **5.01** | **6.02** |
| **Ours** *vs.* LDS | 2.58% | 1.51% | 3.97% | 6.55% | 2.39% | 2.43% | 9.15% | 10.78% |
| **Ours** *vs.* FDS | 3.07% | 3.14% | 4.57% | 5.47% | 5.34% | 4.85% | 6.78% | 6.56% |
| **Ours** *vs.* RankSim | 1.62% | 1.70% | 2.24% | 3.49% | 2.45% | 3.31% | 8.14% | 7.79% |

Table 2: **Main results on IMDB-WIKI-DIR benchmark**.

| Metrics | MAE↓ | | | | GM↓ | | | |
|---|---|---|---|---|---|---|---|---|
| Methods/Shots | All | Many | Median | Few | All | Many | Median | Few |
| ConR-only (Ours) | 7.33 | 6.75 | 11.99 | 22.22 | 4.02 | 3.79 | 6.98 | 12.95 |
| LDS | 7.83 | 7.31 | 12.43 | 22.51 | 4.42 | 4.19 | 7.00 | 13.94 |
| LDS + **ConR** (Ours) | **7.43** | **6.84** | **12.38** | **21.98** | **4.06** | **3.94** | **6.83** | **12.89** |
| FDS | 7.83 | 7.23 | 12.60 | 22.37 | 4.42 | 4.20 | 6.93 | 13.48 |
| FDS + **ConR** (Ours) | **7.29** | **6.90** | **12.01** | **21.72** | **4.02** | **3.83** | **6.71** | **12.59** |
| RankSim | 7.42 | 6.84 | 12.12 | 22.13 | 4.10 | 3.87 | 6.74 | 12.78 |
| RankSim + **ConR** (Ours) | **7.33** | **6.69** | **11.87** | **21.53** | **3.99** | **3.81** | **6.66** | **12.62** |
| LDS + FDS | 7.78 | 7.20 | 12.61 | 22.19 | 4.37 | 4.12 | 7.39 | **12.61** |
| LDS + FDS + **ConR** (Ours) | **7.37** | **7.00** | **12.31** | **21.81** | **4.07** | **3.96** | **6.88** | 12.86 |
| LDS + RankSim | 7.57 | 7.00 | 12.16 | 22.44 | 4.23 | 4.00 | 6.81 | 13.23 |
| LDS + RankSim + **ConR** (Ours) | **7.48** | **6.79** | **12.03** | **22.31** | **4.04** | **3.86** | **6.77** | **12.80** |
| FDS + RankSim | 7.50 | 6.93 | 12.09 | 21.68 | 4.19 | 3.97 | 6.65 | 13.28 |
| FDS + RankSim+ **ConR** (Ours) | **7.44** | **6.78** | **11.79** | **21.32** | **4.05** | **3.88** | **6.53** | **12.67** |
| LDS + FDS + RankSim | 7.69 | 7.13 | 12.30 | 21.43 | 4.34 | 4.13 | 6.72 | **12.48** |
| LDS + FDS + RankSim + **ConR** (Ours) | **7.46** | **6.98** | **12.25** | **21.39** | **4.19** | **4.01** | **6.75** | 12.54 |
| **Ours** *vs.* LDS | 3.67% | 3.54% | 1.07% | 1.21% | 5.75% | 4.07% | 2.37% | 2.08% |
| **Ours** *vs.* FDS | 4.00% | 2.91% | 2.49% | 1.62% | 5.68% | 4.47% | 2.86 % | 2.18% |
| **Ours** *vs.* RankSim | 1.55% | 2.37% | 1.51% | 1.29% | 3.50% | 2.56% | 0.79% | 2.16% |

Table 3: **Main results on NYUD2-DIR benchmark**.

| Metrics | RMSE↓ | | | | $\delta_1$↑ | | | |
|---|---|---|---|---|---|---|---|---|
| Methods/Shots | All | Many | Median | Few | All | Many | Median | Few |
| ConR-only (Ours) | 1.304 | 0.682 | 0.889 | 1.885 | 0.675 | 0.699 | 0.753 | 0.648 |
| FDS | 1.442 | 0.615 | 0.940 | 2.059 | 0.681 | 0.695 | 0.695 | 0.596 |
| FDS + **ConR** (Ours) | **1.299** | **0.613** | **0.836** | **1.825** | **0.696** | **0.800** | **0.819** | **0.701** |
| LDS | 1.387 | **0.671** | 0.913 | 1.954 | 0.672 | **0.701** | 0.706 | 0.630 |
| LDS + **ConR** (Ours) | **1.323** | 0.786 | **0.823** | **1.852** | **0.700** | 0.632 | **0.827** | **0.702** |
| RankSim | - | - | - | - | - | - | - | - |
| Balanced MSE (BNI) | 1.283 | 0.787 | 0.870 | 1.736 | 0.694 | 0.622 | 0.806 | **0.723** |
| Balanced MSE (BNI) + **ConR** (Ours) | **1.265** | **0.772** | **0.809** | **1.689** | **0.705** | **0.631** | **0.832** | 0.698 |
| Balanced MSE (BNI) + LDS | 1.319 | 0.810 | 0.920 | 1.820 | 0.681 | 0.601 | 0.695 | 0.648 |
| Balanced MSE (BNI) + LDS + **ConR** (Ours) | **1.271** | **0.723** | **0.893** | **1.667** | **0.699** | **0.652** | **0.761** | **0.752** |
| **Ours** *vs.* LDS | 4.13% | 4.25% | 10.41% | 6.81% | 3.41% | 0.07% | 13.31% | 13.74% |
| **Ours** *vs.* FDS | 9.92% | 0.03% | 11.06% | 11.37% | 2.20% | 5.27% | 17.84% | 17.62% |
| **Ours** *vs.* Balanced MSE (BNI) | 2.52% | 6.33% | 8.99% | 18.42% | 2.11% | 4.97% | 6.36% | 6.30% |

**Time consumption Analysis.** Table 5 provides the time consumption of **ConR** in comparison to other baselines and VANILLA for the age estimation and depth estimation tasks. VANILLA is a regression model with no technique for imbalanced learning. The reported time consumptions are

Table 4: **Main results of on MPIIGaze-DIR benchmark.**

| Metric | Mean angular error (degrees)↓ | | | |
|---|---|---|---|---|
| **Method/Shot** | **All** | **Many** | **Median** | **Few** |
| **ConR**-only (**Ours**) | 6.16 | 5.73 | 6.85 | 6.17 |
| LDS | 6.48 | 5.93 | 7.28 | 6.83 |
| LDS + **ConR** (**Ours**) | **6.08** | **5.76** | **6.81** | **5.98** |
| FDS | 6.71 | 6.01 | 7.50 | 6.64 |
| FDS + **ConR** (**Ours**) | **6.32** | **5.69** | **6.56** | **5.65** |
| Balanced MSE (BNI) | 5.73 | 6.34 | 6.41 | 5.36 |
| Balanced MSE (BNI)+ **ConR** (**Ours**) | **5.63** | **6.02** | **6.17** | **5.21** |
| **Ours** vs. LDS | 6.17 % | 2.87% | 6.46% | 12.45 % |
| **Ours** vs. FDS | 5.81 % | 5.32% | 12.53% | 14.91 % |
| **Ours** vs. Balanced MSE | 1.75 % | 5.05% | 3.75% | 2.80 % |

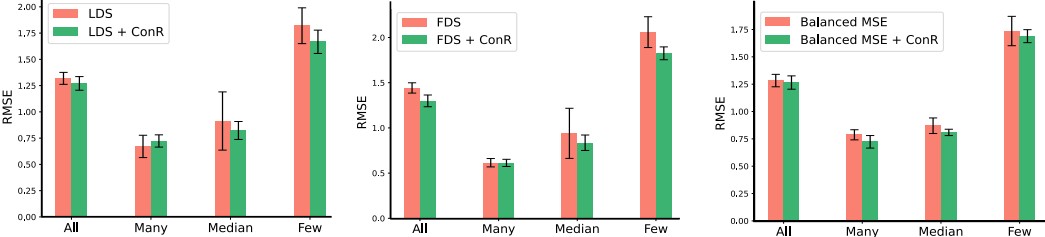

Figure 3: comparison on RMSE results by adding **ConR** on top of the baselines for NYUD2-DIR benchmark.

expressed in seconds for AgeDB-DIR and in minutes for NYUD2-DIR, representing the average forward pass and training time, and were measured using four NVIDIA GeForce GTX 1080 Ti GPUs. Table 5's findings demonstrate that even with a high-dimensional label space, ConR's training time is considerably lower than FDS while remaining comparable to the time complexity of LDS, RankSim, and Balanced MSE. This indicates that ConR is an efficient alternative to FDS without compromising efficiency compared to other well-established methods. This result highlights ConR's ability to handle complex tasks without introducing significant computational overhead.

Table 5: **Time consumption for AgeDB-DIR and NYU2D-DIR benchmarks.**

| Benchmark | AgeDB-DIR | | NYUD2-DIR | |
|---|---|---|---|---|
| **Method/Metric** | **Forward pass (s)** | **Training time (s)** | **Forward pass (m)** | **Training time (m)** |
| VANILLA | 12.2 | 31.5 | 7.5 | 28.3 |
| LDS | 12.7 | 34.3 | 7.9 | 30.2 |
| FDS | 38.4 | 60.5 | 10.8 | 66.3 |
| RankSim | 16.8 | 38.8 | - | - |
| Balanced MSE | 16.2 | 36.3 | 7.9 | 29.4 |
| **ConR** (Ours) | 15.7 | 33.4 | 8.1 | 30.4 |

**Feature visualizations.** We evaluate **ConR** by comparing its learned representations with VANILLA, FDS, and LDS. Using t-SNE visualization, we map ResNet-50's features to a 2D space for the AgeDB-DIR dataset. Fig. 4 demonstrates the feature-label semantic correspondence exploited by **ConR**, compared to VANILLA, FDS and LDS. VANILLA fails to effectively model the feature space regarding three key observations: a) high occurrences of collapses: features of minority samples are considerably collapsed to the majority ones. b) low relative spread: contradicting the linearity in the age estimation task's label space, the learned representation exhibits low feature variance across the label spread (across the colour spectrum) compared to the variance across a single label (same colour). c) Noticeable gaps within the feature space: contradicts the intended continuity in regression tasks. Compared to VANILLA, FDS and LDS slightly alleviate the semantic confusion in feature space. However, as shown in Fig. 4d, **ConR** learns a considerably more effective feature space with fewer collapses, higher relative spread and semantic continuity.

## 4.2 ABLATION STUDIES ON DESIGN MODULES

**Negative sampling, pushing weight and pushing power analysis.** Here we assess the significance of our method's contributions through the evaluation of several variations of **ConR** and investigating the impact of different choices of the hyperparameters in **ConR**. We define four versions of **ConR**: **Contrastive-ConR**: contrastive regularizer where negative peers are selected only based on label

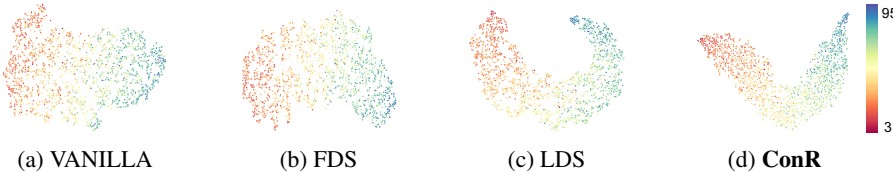

(a) VANILLA      (b) FDS      (c) LDS      (d) **ConR**

Figure 4: Feature visualization on AgeDB-DIR for (a) VANILLA, (b) FDS, (c) LDS and (d) **ConR**.

similarities. **ConR-$\mathcal{S}$**: **ConR** with **no** pushing power assigned to the negative pairs. **ConR-$\eta$**: **ConR** with pushing powers that **are not** proportionate to the instance weights. **ConR-$Sim$**: **ConR** with pushing powers that **are not** proportionate to the label similarities. Table 6 describes the comparison between these three versions on the AgeDB-DIR benchmark and shows the crucial role of each component in deep imbalanced regression. **Contrastive-ConR** is the continuous version of SupCon (Khosla et al., 2020b) that is biased to majority samples (Kang et al., 2021b). Thus, **Contrastive-ConR** shows better results on *many* shot that is due to the higher performance for majority samples, while it degrades the performance for minority ones. However, **ConR** results in a more balanced performance with significant improvement for the minoirty shots.

Table 6: **Ablation results on AgeDB-DIR benchmark.**

| Metrics | MAE↓ | | | |
|---|---|---|---|---|
| Methods/Shots | All | Many | Median | Few |
| **Contrastive-ConR** | 7.69 | **6.12** | 8.73 | 12.42 |
| **ConR-$\mathcal{S}$** | 7.51 | 6.49 | 8.23 | 10.86 |
| **ConR-$Sim$** | 7.54 | 6.43 | 8.19 | 10.69 |
| **ConR-$\eta$** | 7.52 | 6.55 | 8.11 | 10.51 |
| **ConR** | **7.48** | 6.53 | **8.03** | **10.42** |

**Similarity threshold analysis.** We investigate the choice of similarity threshold $\omega$ by exploring the learned features and model performance employing different values for the similarity threshold. Fig. 5a and Fig. 5b compare the feature space learnt with similarity threshold $\omega = 2$ and $\omega = 1$ for **ConR** on the AgeDB-DIR benchmark. **ConR** implicitly enforces feature smoothness and linearity in feature space. A high threshold ($\omega = 2$) is prone to encouraging feature smoothing in a limited label range and bias towards majority samples. As illustrated in Fig. 5b, choosing a lower similarity threshold leads to smoother and more linear feature space. Fig. 5c demonstrates the ablation results for the similarity threshold on the Age-DB-DIR dataset. For AgeDB-DIR, $\omega = 1$ produces the best performance. An intuitive explanation could be the higher thresholds will impose sharing feature statistics to an extent that is not in correspondence with the heterogeneous dynamics in the feature space. For example, in the task of facial age estimation, the ageing patterns of teenagers are different from people in their 20s. For more details on this study please refer to Appendix A.6.

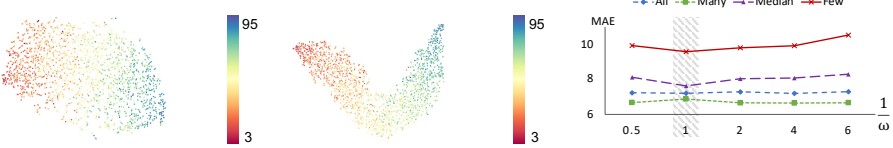

(a) Visualization for $\frac{1}{\omega} = 0.5$    (b) Visualization for $\frac{1}{\omega} = 1$    (c) Similarity threshold analysis.

Figure 5: **Ablation study on the similarity threshold** $\omega$. (a) and (b) compares the learnt feature space for similarity threshold of 2 and 1 respectively. (c) Comparison of different choices of $\omega$ in terms of MAE.

## 5   CONCLUSION

In this work, we propose **ConR**, a novel regularizer for DIR that incorporates continuity to contrastive learning and implicitly encourages preserving local and global semantic relations in the feature space without assumptions about inter-label dependencies. The novel anchor selection proposed in this work consistently derives a balanced training focus across the imbalanced training distribution. **ConR** is orthogonal to all regression models. Our experiments on uni- and multi-dimensional DIR benchmarks show that regularizing a regression model with **ConR** considerably lifts its performance, especially for the under-represented samples and high-dimensional label spaces. **ConR** opens a new perspective to contrastive regression on imbalanced data.

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

## A    EXPERIMENTS

### A.1    EMPIRICAL ANALYSIS OF THE MOTIVATION

The key motivation of the design of **ConR** is that features of minority samples tend to collapse to their majority neighbours. **ConR** highlights these situations by defining a penalty based on the misalignments between prediction similarities and label similarities. Further, **ConR** regularizes a regression model by minimizing the defined penalty in a contrastive manner.

Here we first define the penalty $P(\mathrm{y})$ for each prediction value $\mathrm{y} \in \mathcal{D}_y$. $P(\mathrm{y})$ denotes the average of the regression errors of the samples with the similar prediction value of y: $P(\mathrm{y}) = \frac{1}{N_{\mathrm{y}}} \sum_{i=0}^{N_{\mathrm{y}}} \mathcal{L}_{\mathcal{R}}(x_i, y_i)$, where $\mathcal{R}(\mathcal{E}(x_i)) \simeq \mathrm{y}$, $y_i \not\simeq \mathrm{y}$ and $N_{\mathrm{y}}$ is the number of the samples collapsed to the feature space of samples labelled with y.

To empirically confirm the motivation behind **ConR**, we investigate the importance of the penalty term in regression tasks. In addition, we show that regularizing a regression model with **ConR** consistently alleviate the penalty term defined by **ConR** and this optimization considerably contributes to imbalanced regression. Fig. 6a and Fig. 6b demonstrate the comparison between LDS and **ConR** in terms of the training loss curve and validation loss curve, respectively. Moreover, Fig. 6c shows the trend of the expected value of $P(\mathrm{y})$ over the label space, throughout the training. Comparing Fig. 6c with Fig. 6a and Fig. 6b, $P(\mathrm{y})$ follows the same decreasing pattern as training loss and validation loss. This observation show that penalizing the penalty term is highly coupled with the learning process. In addition, Fig. 6 shows that **ConR** outperform LDS with a considerable gap, particularly in terms of $P(\mathrm{y})$; showing that **ConR** regularizer consistently alleviates the penalty and significantly contributes to the imbalanced regression.

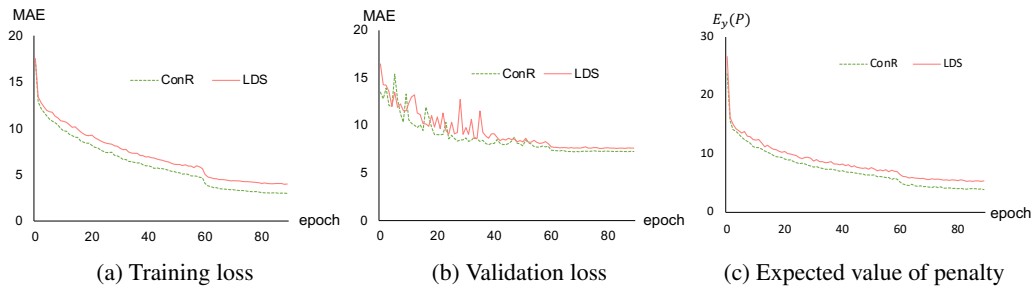

(a) Training loss     (b) Validation loss     (c) Expected value of penalty

Figure 6: Comparison of the performance of LDS and **ConR** on AgeDB-DIR benchmark in terms of (a)training loss, (b)validation loss, and (c)regularizer penalty.

Fig. 7a shows the training label distribution and Fig. 7b depicts the difference of $P(\mathrm{y})$ across the label space between **ConR** and LDS. It empirically shows the considerable improvement of **ConR** over LDS in terms of the defined penalty. More improvement over the majority of samples is intuitive because due to the imbalanced distribution, most of the collapses in feature space happen in the majority areas and decreasing the penalty in these areas contributes the most to the imbalanced regression. Finally, Fig. 7c compares the regression error of **ConR** and LDS and we observe **ConR** results in a considerable improvement over LDS, especially for minority samples.

### A.2    DATASET DETAILS

**Age estimation.**    We evaluated our method on two DIR benchmarks for age estimation curated by Yang et al. (2021): IMDB-WIKI-DIR and AgeDB-DIR. IMDB-WIKI (Rothe et al., 2018) has 191.5K images for training, and 11.0K images for validation and testing, respectively. To structure IMDB-WIKI-DIR, Yang et al. (2021) bin the label space with a bin length of 1 year, where the minimum age is 0 and the maximum age is 186. The bin density varies between 1 and 7,149. The AgeDB dataset (Moschoglou et al., 2017) has 16,488 samples. Yang et al. (2021) constructed AgeDB-DIR in a similar manner as IMDB-WIKI-DIR, where the minimum age is 0 and the maximum age is 101. The maximum number of samples per bin is 353 images and the minimum bin density is 1.

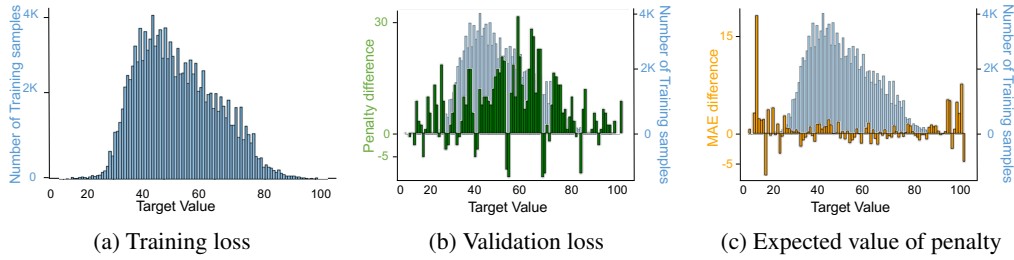

(a) Training loss      (b) Validation loss      (c) Expected value of penalty

Figure 7: Qualitative analysis of **ConR** in terms of regularizer penalty on AgeDB-DIR benchmark. (a) training label distribution. (b) Penalty difference (LDS minus **ConR**) on balanced test data. (c) MAE/ difference (LDS minus **ConR**) on balanced test data.

There are 12,208 training samples and the validation set and test set are made balanced with 2,140 samples each.

**Depth estimation.** Yang et al. (2021) created NYUD2-DIR based on the NYU Depth Dataset V2 Silberman et al. (2012). NYU Depth Dataset V2 has images and corresponding depth maps for different indoor scenes and the task is to predict the depth maps from the RGB scene images. The upper bound of the depth maps is 10 meters and the lower bound is 0.7 meters. Following standard practices. There are 50K images for training and 654 images for testing. Yang et al. (2021) use the bin length of 0.1-meter bin density varies between $1.13 \times 106$ and $1.46 \times 108$. For a balanced test set, Yang et al. (2021) randomly select 9,357 test pixels for each bin from 654 test images with a total of $8.70 \times 105$ test pixels in the NYUD2-DIR test set.

**Gaze estimation.** We used a subset of the MPIIGaze dataset comprising 45,000 training samples from 15 individuals, with 3,000 samples per person. The dataset is naturally imbalanced over the 2D training label distribution.

Here we provide the details of deriving the proposed MPIIGaze-DIR benchmark from the MPIIGaze dataset. the label space of MPIIGaze is 2-dimensional with one dimension ranging from -0.39 to 0.08 and the other from -0.72 to 0.67. Fig. 8a shows the imbalanced distribution across each dimension separately and Fig. 8b shows the joint distribution of the label space with a grid size of 10. Considering the joint distribution, we define thresholds on the bin densities to curate shots (i.e. *many*, *median*, and *few*) as follows: 1300 or more for the *many*-shot, 700 to 1300 for the *median*-shot and less than 700 for the *few*-shot. Next, with the code snippet provided in Fig. 9, We assign the samples to their corresponding shots.

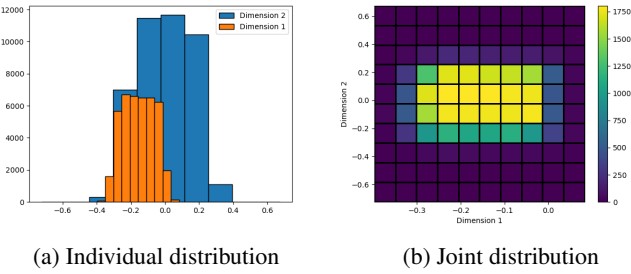

(a) Individual distribution      (b) Joint distribution

Figure 8: Distribution of the 2-dimensional label space of MPIIGaze benchmark.

For training, we follow a leave-one-out scheme where each time one person is dedicated to validation, one person for testing and 13 people for training. The reported results are in terms of average test results among all the 15 people. To create a balanced test set, we take 200 samples from each shot similar to the methods used in the FDS work. Follow the baselines (Gong et al., 2022), only the training data for these tasks is imbalanced; the test dataset is balanced.

```
gaze_hist=plt.hist2d(gazes[:,0],gazes[:,1], bins=10)
shot_th = {

    'median': [700,1300],
    'few': [0,700],
    'many': [1300,float('inf')],
}
shot_dict ={}
for shot,th in shot_th.items():
    indices = np.where((gaze_hist[0]>=th[0]) & (gaze_hist[0]<th[1]))

    shot_dict[shot]=np.column_stack((indices[0], indices[1]))
```

Figure 9: Code snippet for MPIIGaze-DIR creation.

### A.3 BASELINES

**ConR** is orthogonal to state-of-the-art imbalanced learning methods, thus we examine the improvement from **ConR** when added on top of existing methods, which we refer to as baselines for our technique: Label- and Feature-Distribution Smoothing (LDS and FDS) encourage local similarities in label and feature space (Yang et al., 2021). RankSim imposes a matching between the order of similarities in label space with these similarities in feature space (Gong et al., 2022). Balanced MSE encourages a balanced prediction distribution (Ren et al., 2022). To investigate the effect of contrastive learning on deep imbalanced regression, we also regularize using infoNCE (Oord et al., 2018) and contrastive architecture of MoCo (He et al., 2020), MoCo V2 (Chen et al., 2020b). Refer to Appendix A.6 for contrastive analysis. Results for RankSim on depth estimation and gaze estimation are omitted as RankSim is not suitable for these tasks.

### A.4 IMPLEMENTATION DETAILS

We use four NVIDIA GeForce GTX 1080 Ti GPU to train all models. For a fair comparison, we follow (Yang et al., 2021) for all standard train/val/test splits. The rest of this section provides the implementation details and choices of hyperparameters for all three datasets.

**Age estimation.** For AgeDB-DIR benchmark and IMDB-WIKI-DIR benchmark, we use Resnet50 for encoder $\mathcal{E}(\cdot)$ and a one-layer fully connected network for the regression module $\mathcal{R}(\cdot)$. The batch size is 64 and the learning rate is $2.5 * 10^{-4}$ and decreases by $10\times$ at epoch 60 and epoch 80. We use the Adam optimizer with a momentum of 0.9 and a weight decay of 1e-4. Following the baselines (Yang et al., 2021) the loss function for regression $\mathcal{L}_R$ is Mean Absolute Error(MAE). All the models are trained for 90 epochs. The augmentations in the age estimation task are random crop and random horizontal flip.

$\eta_j = (0.01)w_j$ and the similarity function $Sim(\cdot, \cdot)$ is inverse Mean Absolute Error(MAE). To resolve *divide by zero* and *infinite numbers*, a pair of samples with MAE distance $< \frac{1}{\omega}$ are considered similar. Further, the pushing weight $\mathcal{S}_{j,q}$ is defined as: $\mathcal{S}_{j,q} = f_{\mathcal{S}}(\eta_j, \frac{1}{MAE(y_j,y_q)}) = \eta_j MAE(y_j, y_q)$. Finally, the similarity threshold $\omega$ is 1, $\tau = 0.2$, $\alpha = 1$, and $\beta = 4$.

**Depth estimation.** For NYUD2-DIR benchmark, we use ResNet-50-based encoder-decoder architecture (Hu et al., 2019). The output size is $114 \times 152$. The batch size is 32 and the learning rate is $1 * 10^{-4}$. All models are trained for 90 epochs with an Adam optimizer. The momentum of the optimizer is 0.9 and its weight decay is $1e - 4$. Following the baselines (Yang et al., 2021; Hu et al., 2019) the loss function for regression $\mathcal{L}_R$ is root-mean-square(RMSE). The augmentations in the depth estimation task are random rotate, color jitter, and random horizontal flip.

To measure the similarity in label space, first, for each sample $(z_j, y_j)$, we take the average value of the depth map, denoted as $\bar{y}_j$. Then, for each pair of samples, we use the root-mean-square(RMSE) to quantify the similarity of this pair in the label space. If the RMSE$(\bar{y}_i, \bar{y}_j)$ is less than $\frac{1}{\omega}$, sample $i$ and sample $j$ are considered similar and otherwise dissimilar. In addition, the pushing weight $\mathcal{S}_{j,q}$ is defined as: $\mathcal{S}_{j,q} = f_{\mathcal{S}}(\eta_j, \frac{1}{RMSE(y_j,y_q)}) = \eta_j RMSE(y_j, y_q)$. The similarity threshold $\omega$ is 5, $\beta = 0.2$, $\tau = 0.7$ and $\eta_j = (0.2)w_j$.

**Gaze estimation.** We reported our results on a balanced test set, containing 600 samples in total and 200 samples per each "many," "median," and "few" shots. Our results were averaged over five random runs to ensure statistical significance. Each run in the evaluation incorporated a leave-one-out scheme, where we performed 15 runs with a single individual as the designated test set. The final results are the Mean Angle Error (in degrees) for all the individuals. The backbone is LeNet, $\beta = 0.4$, $\alpha = 1, \omega = 1$ . $\eta_j = (0.01)w_j$ and $\mathcal{S}_{j,q} = f_\mathcal{S}(\eta_j, \frac{1}{MAE(y_j, y_q)}) = \eta_j MAE(y_j, y_q)$. The batch size and the base learning rate are 32 and 0.01, respectively. The augmentations in the gaze estimation task are random crop, random resize, and colour jitter.

A.5 PERFORMANCE ANALYSIS

All the experimental results are reported as the averages of 5 random runs.

**More results for age estimation.** For a more extensive empirical confirmation that **ConR** is orthogonal to DIR baselines, Table 7 shows the performance improvements when RRT Yang et al. (2021), Focal-R Yang et al. (2021) and Balanced MSE are regularized by **ConR**.

Table 7: **MAE results of ConR on AgeDB-DIR Benchmark and IMDB-WIKI-DIR benchmark.**

| Metric | MAE↓ | | | | | | | |
|---|---|---|---|---|---|---|---|---|
| Benchmark | AgeDB-DIR | | | | IMDB-WIKI-DIR | | | |
| Methods/Shots | All | Many | Median | Few | All | Many | Median | Few |
| RRT | 7.74 | 6.98 | 8.79 | 11.99 | 7.81 | 7.07 | 14.06 | 25.13 |
| RRT + **ConR** (**Ours**) | **7.53** | **6.79** | **7.60** | **10.30** | **7.41** | **6.89** | **13.20** | **23.30** |
| Focal-R | 7.64 | 6.68 | 9.22 | 13.00 | 7.97 | 7.12 | 15.14 | 26.96 |
| Focal-R + **ConR** (**Ours**) | **7.23** | **6.63** | **8.30** | **11.89** | **7.85** | **7.01** | **14.31** | **25.23** |
| Balabced MSE (GAI) | 7.57 | 7.46 | 8.40 | 10.93 | 8.12 | 7.58 | 12.27 | 23.05 |
| Balabced MSE (GAI) + **ConR** (**Ours**) | **7.22** | **6.71** | **7.99** | **9.88** | **7.84** | **7.20** | **12.09** | **22.20** |
| **Ours** vs. RRT | 2.71 % | 2.72% | 13.54% | 14.10 % | 5.12 % | 2.55% | 6.12% | 7.28 % |
| **Ours** vs. Focal-R | 5.37 % | 0.75% | 9.98% | 8.54 % | 1.51 % | 1.54% | 5.48% | 6.42 % |
| **Ours** vs. Balanced MSE (GAI) | 4.62 % | 10.05% | 4.88% | 9.61 % | 3.45 % | 5.01% | 1.47% | 3.69 % |

**Error Reduction.** Here we show the comparison of the Error reduction resulting from adding **ConR** to the deep imbalanced regression baselines (LDS, FDS and RankSim) for age estimation benchmarks(e.g. AgeDB-DIR and IMDB-WIKI-DIR) and (LDS, FDS and Balanced MSE) for gaze estimation benchmark. Fig. 10, Fig. 11 and Fig. 12 empirically confirm significant performance consistency **ConR** introduces to DIR for AgeDB-DIR, IMDB-WIKI-DIR and MPIIGaze-DIR benchmarks, respectively.

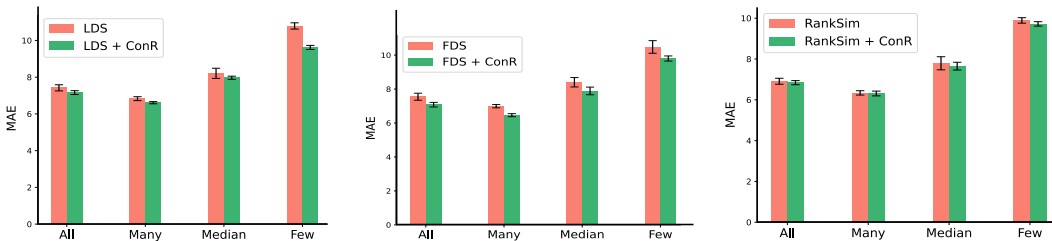

Figure 10: comparison on MAE results by adding **ConR** on top of the baselines for AgeDB-DIR benchmark.

**Feature visualization.** Fig. 13 compares the learned representations by RankSim and Balanced MSE. RanKSim by imposing order relationships encourage high relative spread while Balanced MSE suffer from low relative spread. Both RankSim and Balanced MSE have high occurrences of collapses and noticeable gaps in their feature space. Comparing Fig. 4-d with Fig. 13 shows that **ConR** learns the most effective representations.

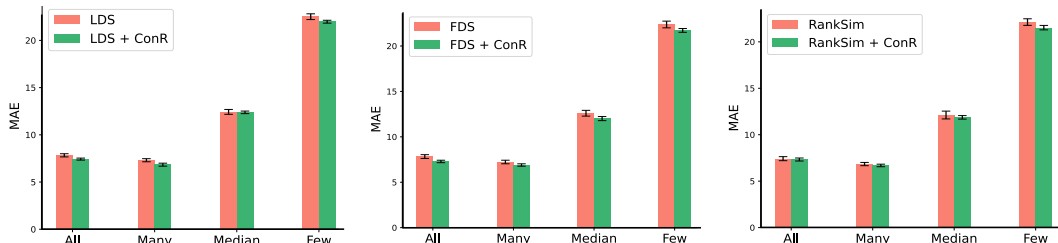

Figure 11: Comparison of the performance gain by regularizing the DIR baselines (LDS, FDS, RankSim) with **ConR** on IMDB-WIKI-DIR benchmark.

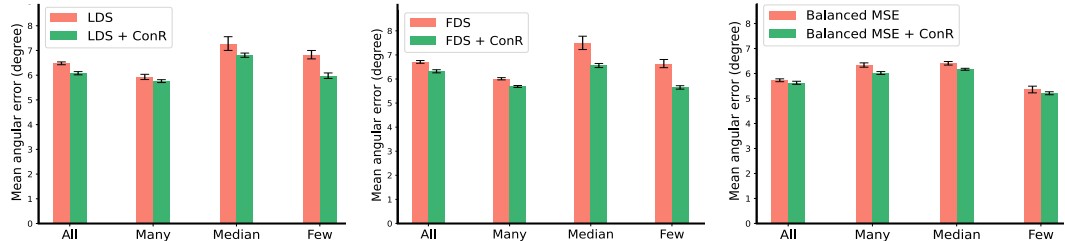

Figure 12: Comparison of the performance gain by regularizing the DIR baselines (LDS, FDS, Balanced MSE) with **ConR** on MPIIGaze-DIR benchmark.

### A.6 ABLATION STUDY

**Negative Sampling, Pushing Weight and Power Analysis.** Table 8, table 9 and table 10 show the significance of the main contributions of **ConR** for IMDB-WIKI-DIR, NYUD2-DIR and MPIIGaze-DIR benchmarks, respectively.

Table 8: **Ablation results of the design modules of ConR on IMDB-WIKI-DIR benchmark.**

| Metric | MAE↓ | | | |
|---|---|---|---|---|
| Method/Shot | All | Many | Median | Few |
| **Contrastive-ConR** | 8.10 | **6.79** | 15.87 | 26.51 |
| **ConR-$\mathcal{S}$** | 7.79 | 6.99 | 14.61 | 25.64 |
| **ConR-$Sim$** | 7.83 | 6.87 | 14.30 | 25.59 |
| **ConR-$\eta$** | **7.76** | 7.10 | 14.25 | 25.33 |
| **ConR (Ours)** | 7.84 | 7.09 | **14.16** | **25.15** |

Table 9: **Ablation results of the design modules of ConR on NYUD2-DIR benchmark.**

| Metric | RMSE↓ | | | |
|---|---|---|---|---|
| Method/Shot | All | Many | Median | Few |
| **Contrastive-ConR** | 1.518 | **0.586** | 1.124 | 2.412 |
| **ConR-$\mathcal{S}$** | 1.410 | 0.670 | 0.941 | 1.954 |
| **ConR-$Sim$** | 1.383 | 0.667 | 0.935 | 1.929 |
| **ConR-$\eta$** | 1.318 | 0.693 | 0.892 | 1.910 |
| **ConR (Ours)** | **1.304** | 0.682 | **0.889** | **1.885** |

**Similarity Threshold Selection.** Fig. 14 shows the ablation study on the similarity threshold for IMDB-WIKI-DIR and NYUD2-DIR benchmarks. $\omega = 1$ and $\omega = 5$ are the best similarity threshold choices for IMDB-WIKI-DIR and NYUD2-DIR benchmarks, respectively.

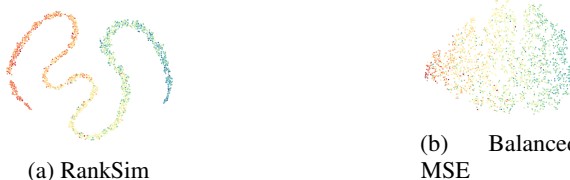

(a) RankSim

(b)     Balanced MSE

Figure 13: Feature visualization on AgeDB-DIR dataset for (a) RankSim, (b) Balanced MSE.

Table 10: **Ablation results of the design modules of ConR on MPIIGaze-DIR benchmark.**

| Metric | Mean Angle Error (degrees)↓ | | | |
|---|---|---|---|---|
| Method/Shot | All | Many | Median | Few |
| **Contrastive-ConR** | 7.11 | **5.00** | 7.73 | 9.89 |
| **ConR-$\mathcal{S}$** | 6.47 | 5.94 | 6.91 | 6.71 |
| **ConR-$Sim$** | 6.39 | 5.69 | 7.09 | 6.48 |
| **ConR-$\eta$** | 6.24 | 5.87 | 6.96 | 6.27 |
| **ConR (Ours)** | **6.16** | 5.73 | **6.85** | **6.17** |

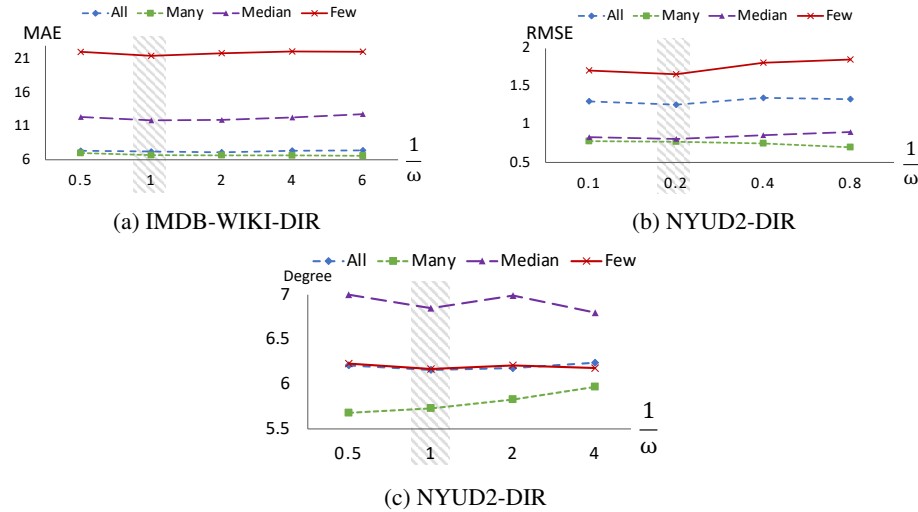

(a) IMDB-WIKI-DIR

(b) NYUD2-DIR

(c) NYUD2-DIR

Figure 14: Comparison of different choices of $\omega$ (a) in terms of MAE for IMDB-WIKI-DIR benchmark, (b) in terms of RMSE for NYUD2-DIR and (c) in terms of Mean angular error (degrees) for MPIIGaze-DIR.

**Hyperparameter selection.** Here we present the selection process of hyperparameters $\alpha$, $\beta$ and $\eta$ for all the benchmarks. Table 11, Table 12, Table 13 and Table 14 show the ablation study of **ConR** on $\alpha$ and $\beta$ in Eq. 4 for AgeDB-DIR, IMDB-WIKI-DIR, NYUD2-DIR and MPIIGaze-DIR benchmarks, respectively. In these tables, hyperparameter $\eta$ is set to the values mentioned in A.4. Additionally, Table 15, Table 16, Table 17 and Table 18 show the ablation study of **ConR** on $\eta$ in Eq. 2 for AgeDB-DIR, IMDB-WIKI-DIR, NYUD2-DIR and MPIIGaze-DIR benchmarks, respectively. In these tables, hyperparameter $\eta$ is set to the values mentioned in A.4.

**Contrastive Regression.** To evaluate the impact of contrastive learning on deep imbalanced regression, we use two contrastive regularizers: **MoCo**: We regularize the baselines with infoNCE loss, using the architecture of with MoCo V1 (He et al., 2020), MoCo V2 (Chen et al., 2020b), and **ConR**. Here we regularized a regression model with both Moco v1 and MoCo v2. Our experiments shows that MoCo v2 degrades the regression performance in some cases. Table 19 compares the performance of a regression model on AgeDB-DIR, IMDB-WIKI-DIR and NYUD2-DIR benchmarks when it is regularized in a contrastive manner with MoCo V1, MoCo V2, and **ConR**. The results

Table 11: **Ablation study on $\alpha$ and $\beta$ in Eq. 4 for AgeDB-DIR benchmark**

| $\alpha$ | $\beta$ | MAE↓ | | | |
|---|---|---|---|---|---|
| | | All | Many | Median | Few |
| 0.5 | 1 | 7.31 | 6.53 | 8.08 | 10.51 |
| 1 | 0.5 | 7.30 | 6.59 | 8.10 | 10.46 |
| 1 | 1 | 7.31 | 6.55 | 8.11 | 10.48 |
| 1 | 2 | 7.35 | 6.58 | 8.07 | 10.46 |
| **1** | **4** | 7.28 | 6.53 | 8.03 | 10.42 |
| 1 | 5 | 7.25 | 6.47 | 8.12 | 10.54 |

Table 12: **Ablation study on $\alpha$ and $\beta$ in Eq. 4 for IMDB-WIKI-DIR benchmark**

| $\alpha$ | $\beta$ | MAE↓ | | | |
|---|---|---|---|---|---|
| | | All | Many | Median | Few |
| 0.5 | 1 | 7.98 | 7.11 | 14.26 | 25.19 |
| 1 | 0.5 | 7.90 | 7.17 | 14.34 | 25.21 |
| 1 | 1 | 7.94 | 7.29 | 14.28 | 25.19 |
| 1 | 2 | 7.80 | 7.21 | 14.23 | 25.17 |
| **1** | **4** | 7.84 | 7.09 | 14.16 | 25.15 |
| 1 | 5 | 7.98 | 7.25 | 14.24 | 25.13 |

Table 13: **Ablation study on $\alpha$ and $\beta$ in Eq. 4 for NYUD2-DIR benchmark**

| $\alpha$ | $\beta$ | RMSE↓ | | | |
|---|---|---|---|---|---|
| | | All | Many | Median | Few |
| 0.5 | 1 | 1.344 | 0.722 | 0.897 | 1.918 |
| 1 | 0.5 | 1.324 | 0.712 | 0.891 | 1.945 |
| **1** | **0.2** | 1.304 | 0.682 | 0.889 | 1.885 |
| 1 | 0.4 | 1.316 | 0.673 | 0.909 | 1.905 |

Table 14: **Ablation study on $\alpha$ and $\beta$ in Eq. 4 for MPIIGaze-DIR benchmark**

| $\alpha$ | $\beta$ | Mean Angle Error (degrees)↓ | | | |
|---|---|---|---|---|---|
| | | All | Many | Median | Few |
| 0.5 | 1 | 6.18 | 5.79 | 6.89 | 6.19 |
| 1 | 0.2 | 6.14 | 5.85 | 6.84 | 6.22 |
| **1** | **0.4** | 6.16 | 5.73 | 6.85 | 6.17 |
| 1 | 1 | 6.09 | 5.64 | 7.05 | 6.24 |

Table 15: **Ablation study of $\eta$ (pushing power) for AgeDB-DIR benchmark.**

| $\eta$ | MAE↓ | | | |
|---|---|---|---|---|
| | All | Many | Median | Few |
| 0.009 | 7.31 | 6.63 | 7.99 | 10.5 |
| **0.01** | 7.28 | 6.53 | 8.03 | 10.42 |
| 0.05 | 7.36 | 6.51 | 8.11 | 10.51 |
| 0.1 | 7.38 | 6.49 | 8.09 | 10.48 |

are reported in terms of MAE for AgeDB-DIR benchmark and IMDB-WIKI-DIR benchmark and in terms of RMSE for NYUD2-DIR dataset. Moco considerably boost the performance of VANILLA and shows that contrastive training significantly improves the regression performance, especially for minority samples. **ConR** incorporate unbiased supervision into the contrastive regression and significantly boost the performance on minority samples with no harm to the learning process for majority samples. As shown in Fig. 15, Moco and **ConR** provide more consistent performance compared to the baseline. In addition, Moco is consistently outperformed by **ConR** and empirically

Table 16: **Ablation study of $\eta$ (pushing power) for IMDB-WIKI-DIR benchmark.**

| $\eta$ | MAE↓ | | | |
|---|---|---|---|---|
| | **All** | **Many** | **Median** | **Few** |
| 0.009 | 7.88 | 7.07 | 14.25 | 25.16 |
| **0.01** | 7.84 | 7.09 | 14.16 | 25.15 |
| 0.05 | 7.88 | 6.99 | 14.20 | 25.16 |
| 0.1 | 7.91 | 7.14 | 14.18 | 25.21 |

Table 17: **Ablation study of $\eta$ (pushing power) for NYUD2-DIR benchmark.**

| $\eta$ | RMSE↓ | | | |
|---|---|---|---|---|
| | **All** | **Many** | **Median** | **Few** |
| 0.1 | 1.312 | 0.674 | 0.886 | 1.891 |
| **0.2** | 1.304 | 0.682 | 0.889 | 1.885 |
| 0.4 | 1.295 | 0.619 | 0.92 | 1.911 |

Table 18: **Ablation study of $\eta$ (pushing power) for MPIIGaze-DIR benchmark.**

| $\eta$ | Mean angular error (degrees)↓ | | | |
|---|---|---|---|---|
| | **All** | **Many** | **Median** | **Few** |
| 0.5 | 6.15 | 5.77 | 6.83 | 6.21 |
| 1 | 6.16 | 5.73 | 6.85 | 6.17 |
| 2 | 6.18 | 5.74 | 6.89 | 6.23 |

confirms **ConR** improves the self-supervised contrastive regularizer by incorporating supervision in an unbiased manner.

Table 19: **Results of contrastive learning analysis on AgeDB-DIR, IMDB-WIKI-DIR and NYUD2-DIR benchmarks.** Results are reported for the whole test data (*all*) and three other shots: *many*, *median*, *few*. At the bottom of the table the improvements of **ConR** with respect to Moco are reported in green for each benchmark, shot and metric. In each column, the best result is in **bold**.

| Benchmark | AgeDB-DIR | | | | IMDB-WIKI-DIR | | | | NYUD2-DIR | | | |
|---|---|---|---|---|---|---|---|---|---|---|---|---|
| Metric | MAE↓ | | | | MAE↓ | | | | RMSE↓ | | | |
| Method \ Shot | All | Many | Median | Few | All | Many | Median | Few | All | Many | Median | Few |
| VANILLA | 7.35 | 6.56 | 8.23 | 12.37 | 8.06 | 7.23 | 15.12 | 26.33 | 1.477 | **0.591** | 0.952 | 2.123 |
| + Moco V1 | 7.33 | **6.50** | 8.19 | 11.72 | 7.89 | 7.13 | 14.78 | 26.11 | 1.370 | 0.601 | 0.902 | 1.912 |
| + Moco V2 | 7.47 | 6.21 | 8.75 | 12.75 | 8.12 | 6.99 | 15.02 | 26.01 | 1.404 | 0.632 | 0.978 | 2.207 |
| + ConR | **7.28** | 6.53 | **8.03** | **10.42** | **7.84** | **7.09** | **14.16** | **25.15** | **1.304** | 0.682 | **0.889** | **1.885** |
| **ConR** *vs.* Moco | 0.68% | -0.46% | 1.96% | 11.10% | 0.63% | 0.56% | 4.20% | 3.68% | 4.82% | -1.63% | 1.44% | 4.41% |

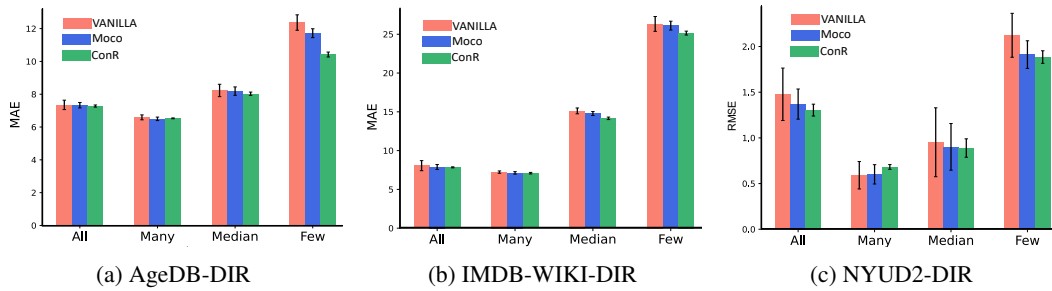

(a) AgeDB-DIR      (b) IMDB-WIKI-DIR      (c) NYUD2-DIR

Figure 15: Comparison of the performance gain of regularizing VANILLA regression model with contrastive regularizers: Moco V1 and **ConR** on (a) AgeDB-DIR, (b) IMDB-WIKI-DIR anf (c) NYUD2-DIR benchmarks. It shows contrastive regularizer consistently lifts the performance of the baseline, particularly on minority samples.

## B MORE THEORETICAL INSIGHTS

Here we theoretically justify the effectiveness of **ConR** by deriving a upper bound on the probability of incorrect labelling of minority samples. We show that minimizing $\mathcal{L}_{ConR}$ robustly minimizes this probability of mislabeling for minority samples and consequently improves the generalizability.

In the following, we'll derive the upper bound: following (Oord et al., 2018; Wang et al., 2022) we define the density ratio $f(\hat{y}, x)$ in InfoNCE Oord et al. (2018) to be $\frac{p(\hat{y}|x)}{p(\hat{y})}$ (Oord et al., 2018; Wang et al., 2022) that is estimated by $\exp(z_j \cdot z_i/\tau)$ in Eq. 1 like other contrastive objective functions (He et al., 2020; Chen et al., 2020a). $p(\hat{y}|x)$ is the desired prediction distribution and $\hat{Y}_j = (\hat{y}_j - \omega, \hat{y}_j + \omega)$.

For each anchor $x_j$ the $\mathcal{L}_{ConRj}$ in Eq. 1 is defined to be:

$$\mathcal{L}_{ConRj} = -\log \frac{1}{N_j^+} \sum_{z_i \in K_j^+} \frac{\exp(z_j \cdot z_i/\tau)}{\sum_{z_p \in K_j^+} \exp(z_j \cdot z_p/\tau) + \sum_{z_q \in K_j^-} \mathcal{S}_{j,q} \exp(z_j \cdot z_q/\tau)} \quad (6)$$

Then, $\mathcal{L}_{ConRj}$ can be rewritten as:

$$\mathcal{L}_{ConRj} = -\log \frac{1}{N_j^+} \sum_{i=0}^{N_j^+} \frac{f(\hat{y}_j, x_i)}{\sum_{p=0}^{K_j^+} f(\hat{y}_j, x_p) + \sum_{q=0}^{K_j^-} \mathcal{S}_{j,q} f(\hat{Y}_j, x_q)} \quad (7)$$

Following Jensen's inequality we have:

$$\mathcal{L}_{ConRj} = -\log \frac{1}{N_j^+} \sum_{i=0}^{N_j^+} \frac{f(\hat{y}_j, x_i)}{\sum_{p=0}^{N_j^+} f(\hat{y}_j, x_p) + \sum_{q=0}^{K_j^-} \mathcal{S}_{j,q} f(\hat{Y}_j, x_q)} \quad (8)$$

$$\overset{\text{McShane (1937)}}{\geq} -\frac{1}{N_j^+} \sum_{i=0}^{N_j^+} \log \frac{f(\hat{y}_j, x_i)}{\sum_{p=0}^{K_j^+} f(\hat{y}_j, x_p) + \sum_{q=0}^{N_j^-} \mathcal{S}_{j,q} f(\hat{Y}_j, x_q)} \quad (9)$$

where $x_q$ are the negative sample, and $\hat{y}_j$ is the prediction of positive samples. prediction of $x_q$ mistakenly fall in the range of $(\hat{y}_j \pm \omega)$. Further, we have:

$$\mathcal{L}_{ConRj} \geq -\frac{1}{N_j^+} \sum_{i=0}^{K_j^+} \log \frac{f(\hat{y}_j, x_i)}{\sum_{p=0}^{N_j^+} f(\hat{y}_j, x_p) + \sum_{q=0}^{N_j^-} \mathcal{S}_{j,q} f(\hat{Y}_j, x_q)} \quad (10)$$

$$= \frac{1}{N_j^+} \sum_{i=0}^{N_j^+} \log \left[ \frac{\sum_{p=0}^{N_j^+} f(\hat{y}_j, x_p)}{f(\hat{y}_j, x_i)} + \frac{\sum_{q=0}^{N_j^-} \mathcal{S}_{j,q} f(\hat{Y}_j, x_q)}{f(\hat{y}_j, x_i)} \right] \quad (11)$$

$$\geq \frac{1}{N_j^+} \sum_{i=0}^{N_j^+} \log \left[ \frac{\sum_{p=0}^{N_j^+} f(\hat{y}_j, x_p)}{\sum_{i=0}^{N_j^+} f(\hat{y}_j, x_i)} + \frac{\sum_{q=0}^{K_j^-} \mathcal{S}_{j,q} f(\hat{Y}_j, x_q)}{f(\hat{y}_j, x_i)} \right] \quad (12)$$

$$\geq \frac{1}{N_j^+} \sum_{i=0}^{N_j^+} \log \left[ 1 + \frac{\sum_{q=0}^{N_j^-} \mathcal{S}_{j,q} f(\hat{Y}_j, x_q)}{f(\hat{y}_j, x_i)} \right] \quad (13)$$

$$\geq \frac{1}{N_j^+} \sum_{i=0}^{N_j^+} \log \left[ \frac{\sum_{q=0}^{N_j^-} \mathcal{S}_{j,q} f(\hat{Y}_j, x_q)}{f(\hat{y}_j, x_i)} \right] \quad (14)$$

$$= \frac{1}{N_j^+} \left[ \log \sum_{q=0}^{N_j^-} \mathcal{S}_{j,q} f(\hat{Y}_j, x_q) - \log \sum_{i=0}^{N_j^+} f(\hat{y}_j, x_i) \right] \quad (15)$$

Next, the $\mathcal{L}_{ConR}$ is:

$$\mathcal{L}_{ConR} = \frac{1}{2N} \sum_{j=0}^{2N} \mathcal{L}_{ConRj} \tag{16}$$

$$\geq \frac{1}{2N} \sum_{j=0}^{2N} \frac{1}{N_j^+} \Big[ \log \sum_{q=0}^{N_j^-} \mathcal{S}_{j,q} f(\hat{Y}_j, x_q) - \log \sum_{i=0}^{N_j^+} f(\hat{y}_j, x_i) \Big] \tag{17}$$

$$\geq \frac{1}{2N} \sum_{j=0}^{2N} \frac{1}{2N} \Big[ \log \sum_{q=0}^{N_j^-} \mathcal{S}_{j,q} f(\hat{Y}_j, x_q) - \log \sum_{i=0}^{N_j^+} f(\hat{y}_j, x_i) \Big] \tag{18}$$

$$= \frac{1}{4N^2} \sum_{j=0}^{2N} \Big[ \log \sum_{q=0}^{N_j^-} \mathcal{S}_{j,q} f(\hat{Y}_j, x_q) - \log \sum_{i=0}^{N_j^+} f(\hat{y}_j, x_i) \Big] \tag{19}$$

$$= \frac{1}{4N^2} \Big( \sum_{j=0}^{2N} \log \sum_{q=0}^{N_j^-} \mathcal{S}_{j,q} f(\hat{Y}_j, x_q) - \sum_{j=0}^{2N} \log \sum_{i=0}^{N_j^+} f(\hat{y}_j, x_i) \Big) \tag{20}$$

$$= \frac{1}{4N^2} \Big( \sum_{j=0}^{2N} \log \sum_{q=0}^{N_j^-} \mathcal{S}_{j,q} f(\hat{Y}_j, x_q) - \sum_{j=0}^{2N} \log \sum_{i=0}^{N_j^+} \frac{p(\hat{y}_j|x_i)}{p(\hat{y}_j)} \Big) \tag{21}$$

$$\overset{\frac{p(\hat{y}_j|x_i)}{p(\hat{y}_j)}<1}{\geq} \frac{1}{4N^2} \Big( \sum_{j=0}^{2N} \log \sum_{q=0}^{N_j^-} \mathcal{S}_{j,q} f(\hat{Y}_j, x_q) - \sum_{j=0}^{2N} \log \sum_{i=0}^{N_j^+} 1 \Big) \tag{22}$$

$$= \frac{1}{4N^2} \Big( \sum_{j=0}^{2N} \log \sum_{q=0}^{N_j^-} \mathcal{S}_{j,q} f(\hat{Y}_j, x_q) - 2N \log N_j^+ \Big). \tag{23}$$

Further, we have:

$$\mathcal{L}_{ConR} = \frac{1}{4N^2} \Big( \sum_{j=0}^{2N} \log \sum_{q=0}^{N_j^-} \mathcal{S}_{j,q} f(\hat{Y}_j, x_q) - 2N \log N_j^+ \Big) \tag{24}$$

$$\geq \frac{1}{4N^2} \Big( \sum_{j=0}^{2N} \sum_{q=0}^{N_j^-} \log \mathcal{S}_{j,q} f(\hat{Y}_j, x_q) - 2N \log N_j^+ \Big) \tag{25}$$

$$= \frac{1}{4N^2} \Big( \sum_{j=0}^{2N} \sum_{q=0}^{N_j^-} \log \mathcal{S}_{j,q} \frac{p(\hat{Y}_j|x_q)}{p(\hat{Y}_j)} - 2N \log N_j^+ \Big) \tag{26}$$

$$\overset{p(\hat{Y}_j)<1}{\geq} \frac{1}{4N^2} \Big( \sum_{j=0}^{2N} \sum_{q=0}^{N_j^-} \log \mathcal{S}_{j,q} p(\hat{Y}_j|x_q) - 2N \log N_j^+ \Big) \tag{27}$$

$$\geq \frac{1}{4N^2} \Big( \sum_{j=0}^{2N} \sum_{q=0}^{N_j^-} \log \mathcal{S}_{j,q} p(\hat{Y}_j|x_q) - 2N \log(2N) \Big). \tag{28}$$

Then:

$$\frac{1}{4N^2} \sum_{j=0}^{2N} \sum_{q=0}^{N_j^-} \log \mathcal{S}_{j,q} p(\hat{Y}_j|x_q) \leq \mathcal{L}_{ConR} + \frac{log(2N)}{2N}. \tag{29}$$

As explained in 3, among the 2N augmented samples only the ones with the confusion around them are chosen as anchors. Assuming $A$ is the set of selected anchors that is a subset of the 2N augmented samples we have:

$$\frac{1}{4N^2} \sum_{j=0,x_j \in A}^{2N} \sum_{q=0}^{K_j^-} \log \mathcal{S}_{j,q} p(\hat{Y}_j|x_q) \leq \mathcal{L}_{ConR} + \epsilon, \quad \epsilon \overset{N \to \infty}{\to} 0 \tag{30}$$

In Eq. 30, $p(\hat{Y}_j|x_q)$ is the likelihood of sample $x_q$ with an incorrect prediction $y_q \in \hat{Y}_j$. We refer to $p(\hat{Y}_j|x_q)$ as the probability of collapse for $x_q$. The left-hand side presents this probability for all the negative pairs.

Regarding the empirical study by Yang et al. (2021), when learning a regression function from the imbalanced data, the representations of minority samples tend to collapse to the majority ones. Since in the definition of $\mathcal{L}_{ConRj}$ in Eq. 1, $x_q$ show the collapsed minority samples, minimizing the left side of the inequality in Eq. 30 is the intended optimization in deep imbalanced regression.

The left-hand side is the probability of all collapses during the training and regarding Eq. 30 Convergence of $\mathcal{L}_{ConR}$ tightens the upper bound for it. Minimizing the left-hand side can be explained with two non-disjoint scenarios: either the number of anchors or the degree of collapses is reduced. Here the degree of collapse for each negative sample refers to the quantified disagreement between the label similarity and prediction similarity as discussed in section 3. In addition, each collapse probability is weighted with $\mathcal{S}_{j,q}$, leading to penalizing the incorrect predictions with regard to their severity. In other words, **ConR** penalizes the bias probabilities with a re-weighting scheme, where the weights are defined based on the agreement of the predictions and labels.

## C  ALGORITHM OF **ConR**

Algorithm 1 shows the pseudo-code of regularizing a regression model using **ConR**. In this algorithm, $\mathcal{S}_{j,q}$ is the pushing weights for selected anchor $z_j$ and its negative pair $z_q$ (Eq. 2). $K_j^+$ and $K_j^-$ are positive pairs of $z_j$, and negative pairs of $z_j$, respectively.

---

**Algorithm 1 ConR**: Contrastive regularizer for deep imbalanced regression

---

**Require:**

    input samples $\{(x_i, y_i)\}_{i=0}^{N}$

    feature encoder $\mathcal{E}(.)$

    regression function $\mathcal{R}(.)$

  **for** e in epochs **do**

      **for** $b = \{(x_i, y_i)\}_{i=0}^{N_b}$ in Batches **do**

        $\{x_j^a, y_j\}_{j=0}^{2N_b} \overset{\text{Augmentations}}{\leftarrow} \{x_i, y_i\}_{i=0}^{N_b}$

        $\{z_j\}_{j=0}^{2N} \leftarrow \mathcal{E}(\{x_j^a\}_{j=0}^{2N_b})$              $\triangleright z_j$ : feature embeddings of augmented input $x_j^a$.

        $\{\hat{y}_j\}_{j=0}^{2N_b} \leftarrow \mathcal{R}(\mathcal{E}(\{z_j\}_{j=0}^{2N_b}))$             $\triangleright \hat{y}_j$ : prediction of $z_j$.

        $\{K_j^+, K_j^-\}_{j=0}^{2N_b} \leftarrow \{z_j, y_j, \hat{y}_j\}_{j=0}^{2N_b}$

        **for** $j \in (0, 2N_b)$ **do**

           **if** $N_j^- == 0$ **then**

               $\mathcal{L}_{ConRj} \leftarrow 0$      $\triangleright$ Samples with no negative pairs are not selected as anchors.

           **else**

               $\{\mathcal{S}_{j,q}\}_{q=0}^{N_j^-} \leftarrow f_\mathcal{S}(y_j, \{y_q\}_{q=0}^{N_j^-})$        $\triangleright \mathcal{S}_{j,q}$: pushing weight (Eq. 2)

               $\mathcal{L}_{ConRj} \leftarrow \text{computeConR}(z_j, K_j^+, K_j^-, \{\mathcal{S}_{j,q}\}_{q=0}^{N_j^-})$      $\triangleright$ Eq. 1

           **end if**

        **end for**

        $\mathcal{L}_{ConR} \overset{\text{Average}}{\leftarrow} \{\mathcal{L}_{ConRj}\}_{j=0}^{2N_b}$                $\triangleright$ Eq. 3

        $\mathcal{L}_\mathcal{R} \leftarrow \text{computeRegressionLoss}(\{\hat{y}_j, y_j\}_{j=0}^{N_b})$

        $L_{sum} \leftarrow \alpha\mathcal{L}_\mathcal{R} + \beta\mathcal{L}_{ConR}$

        $L_{sum}.\text{backPropagate}()$

      **end for**

  **end for**

**Ensure:** Trained regression function $\mathcal{R}(.)$ and trained feature encoder $\mathcal{E}(.)$

---

