# OpenReview forum: "ConR: Contrastive Regularizer for Deep Imbalanced Regression"
_ICLR.cc/2024/Conference — ICLR 2024 poster_

### Official Review · Reviewer_Yj3F · 2023-10-21

**Soundness:** 3 good
**Presentation:** 3 good
**Contribution:** 3 good
**Rating:** 6
**Confidence:** 4

**Summary:**

For imbalanced regression, the authors propose ConR, which uses
supervised contrastive loss as part of the loss function.  They define
Sim(label1,label2) > omega, a threshold hyperparameter, as similar
labels.  For each of the input instances, it generates two augmented
instances.  If the actual labels of two augmented instances are
similar, they form a positive pair.  If the predicted labels of two
instances are similar, but the actual labels are not, they form a
negative pair.  For an augmented instance, a set of positive samples
and a set of negative samples are found.  Augmented instances with at
least one negative sample are called anchors, which participate in
contrastive loss.  For each anchor, the fraction in the regular
contrastive loss is summed over all positive samples.  For the
negative samples in the denominator, they have a "pushing weight" S,
which is a function of the density-based weight of the anchor and the
Sim(anchor_label, negative_sample_label).  L_conR is an average of the
contrastive loss of up to 2N anchors. The overall loss is a weighted
sum of the regular regression loss and L_conR.

ConR was evaluated on 4 datasets, one of which has
multi-dimensional labels, and compared with 4 recent techniques.
Empirical results indicate adding ConR generally improves performance.

**Strengths:**

Imbalanced regression is an interesting problem.
The proposed technique adapts supervised contrastive loss from
classification to regression.  Particularly, they added a pushing
factor for the negative samples based on similarity in labels and density,
which is interesting.  Also, negative samples have not just
different labels but also similar predictions.  Empirical results
indicate adding ConR generally improves performance.

**Weaknesses:**

The proposed contrastive loss, Eq 1, could be further explained and
justified.  Empirical evaluation could be improved by adding "ConR
only" and representations of RankSim and Balanced MSE, which are more
recent techniques.  The similarity threshold omega seems to be an
important parameter, further insights could be explored.

More details are in questions below.

**Questions:**

1. Eq 1, the outermost summation: Summing over i's implies j's with more
   positive samples would have more contribution.  Is that desirable?  If so,
   what is the main reason?

2. L_R is on the original instances, while L_conR is on 2
   augmentations of each of the original instances.  That is, the
   original instances are not directly involved in L_conR--is that
   correct?  If so, what is the main reason?

3. Similarity threshold omega between labels dependent on
   applications and range of the labels (1-100 vs 1-10^6), any
   further insights?  It seems to be trial and error as a
   hyperparameter.

4. How does ConR alone, not inconjunction with another technique,
   perform?  Including it in Tables 1-3, would be beneficial.

5. How do the representations from Balanced MSE and RankSim, which are
   more recent, compare with ConR.  Including them in Fig 4 would be
   important.

6. In Fig 5, why 1/omega, instead of omega, is used?  In the approach
   section, 1/omega was not discussed. "Fig. 5b, choosing a higher
   similarity threshold"--it seems similarity threshold omega is
   smaller at 1 (1/omega is higher).

7. p9: Could you further explain: "sharing feature statistics to an
   extent that is not in correspondence with the heterogeneous
   dynamics in the feature space"?

8. Sec 3.2.1: how are the instances augmented?  It seems to be not
   discussed in the approach or experiments.

Comments:

Sec 3.2.2: for completeness, f_S could have more description (it seems
to be a simple product in the appendix)

minor:

Eq 3: 1/2N assumes all 2N augmented instances are anchors, but some
might not be anchors (if I understand correctly).

Kang et al. 21 and Khosla et al. 20 have duplicated entries in
References.

---

> ### Author Response · Authors · 2023-11-22
> **Response to Reviewer Yj3F - part 1**
>
> Thank you very much for your thorough review and insightful feedbacks. Here we address your valuable comments:
> ## The outermost summation in Eq. 1
> Thank you very much for highlighting this point. In the design of ConR the summation over j’s is divided by the number of positive samples so that for each anchor, the average of contribution of the positive pair is incorporated into ConR. It is mistakenly dropped from Eq. 1, and has been fixed in the updated manuscript.
> ## Input augmentations for regression loss and ConR
> We appreciate your careful examination of the objective functions. Data augmentation is an indispensable component of contrastive learning and also deep visual models because it boosts robustness and generalization by bringing representational invariance to the learned models [1]. Moreover, an essential feature of ConR's design is its inherent ease of integration and orthogonality with the targeted regression baseline for regularization. Whether the regression baseline augments inputs before feeding them to L_R, our approach mirrors this process. Consequently, the incorporation of ConR requires minimal adjustments and can be seamlessly integrated without necessitating substantial considerations.
> Concerning L_R in our evaluation process, in all baselines and for all benchmarks, each input sample undergoes one augmentation before being incorporated into L_R. This aligns with the common practice observed in baseline methodologies. This highlights the straightforward integration of ConR into existing frameworks.\
> ## Similarity Threshold selection
> The determination of the similarity threshold is task-dependent. For instance, in the context of age estimation, employing a yearly measure is intuitively relevant. In our efforts to optimize this parameter, we conducted an ablation study ("Similarity threshold analysis" in section 4.2 of the paper and section A.5 of the appendix) to explore the impact of varying the number of years considered as the distance threshold for defining the similarity threshold. Our ablation study shows that a high similarity threshold leads to bias toward majority samples and a low similarity threshold will lead to disagreements with label relationships. This systematic examination enhances the robustness and applicability of ConR across different tasks.
> ## Performance of ConR-only
> Thank you for highlighting the importance of additional report in the experiment section. In response to your comment, in the updated manuscript we have added the performance of ConR independently, referred to as "ConR-only," in the first row of tables 1-4. This allows for a clear and comprehensive understanding of ConR's effectiveness in isolation, without the influence of additional techniques.
> ## Representations of Balanced MSE and RankSim
> Thank you for your attention. Due to space constraints, we omitted the visualizations of Balanced MSE and RankSim in Figure 4. For the representations of Balanced MSE[2] and RankSim, as well as their respective analyses, please refer to Figure 13 and "Feature visualization" in the updated manuscript appendix.\
> Figure 13 shows that RanKSim by imposing order relationships encourages high relative spread while Balanced MSE suffers from low relative spread. Both RankSim and Balanced MSE have high occurrences of collapses and noticeable gaps in their feature space.  Comparing the representations of Balanced MSE and RankSim with the one of ConR shows that ConR learns the most effective representations.
> ## $1/\omega$ instead of $\omega$
> Figure 5 presents the ablation study on the age estimation task. To facilitate a more intuitive comparison, we employed $1/\omega$ as it represents the distance, and in the context of age estimation, the distance corresponds to age difference. The figure illustrates the optimal choice for the age difference. Moreover, to maintain consistency, we have used this notation for the ablation study on the similarity threshold for other tasks as well.
> ## Figure 5-b: similarity threshold omega is smaller at 1
> Thank you very much for raising this valid concern. It is fixed in the updated manuscript.
>
> ### References
> [1] Cui, Jiequan, et al. "Parametric contrastive learning.", ICCV 2021.\
> [2] Ren et al., "Balanced MSE for Imbalanced Visual Regression", CVPR 2022.

---

> > ### Author Response · Authors · 2023-11-22
> > **Response to Reviewer Yj3F - part 2**
> >
> > ## Clarification on "sharing feature statistics to an extent that is not in correspondence with the heterogeneous dynamics in the feature space"
> > Thank you for your thorough consideration. This implies that choosing a low similarity threshold may result in mistakenly considering samples as similar when they should not be. For instance, in the age estimation task, individuals in the age range of 20-22 might share facial similarities, but they are dissimilar to teenagers. Therefore, the similarity threshold should not be set so low as to consider these two groups as similar.
> > Moreover, a high similarity threshold can also be ineffective, as it may lead to the repulsion of samples that should be considered similar. Striking the right balance in choosing the similarity threshold is crucial to ensuring accurate and meaningful contrastive learning.
> >
> > ## Adopted Augmentations
> > For a fair comparison to the baselines, we used the task-specific augmentations for each benchmark. Please refer to the added details on augmentations in section A.3 of the Appendix in the updated manuscript.
> >
> > ## Description of $f\_\mathcal{S}$
> > We appreciate your attention to the details of the design of ConR. Your observation about a simple product is correct. $f\_\mathcal{S}$ can be designed in a task-specific manner. We took the simple product to keep the method simple.
> >
> > ##  $1/2N$ in Eq. 3
> > That is correct. Eq. 3 measures the average of effect of the collapses in each training step. Thus all augmented samples incorporate to $\mathcal{L}_{ConR}$.
> > For the non-anchor samples, the ${\mathcal{L}\_{ConR}}\_j$ is zero. For the anchors the value of Eq. 1 and for the non-anchors, zeros will be used in Eq. 3.
> >
> > ## Duplicated entries in References
> > Thank you very much for highlighting the duplicated refrecences. In the updated manuscript, the duplicates are fixed.

---

> > > ### Comment · Reviewer_Yj3F · 2023-11-22
> > > **comments on authors' response**
> > >
> > > "ConR only" seems to have better performance than LDS and FDS, but not necessarily RankSim.
> > >
> > > Representation of ConR vs RankSim in Figure 4d and Figure 13a.   ConR seems to have a larger spread for the same/similar age/color than RankSim.
> > >
> > > Further explanation on what  "feature statistics" and "heterogeneous dynamics" are would be helpful.  It seems your response is the selection of similarity threshold omega, which affects how well "positive" and "negative" pairs are selected, and hence the learned feature space.  What are the "feature statistics"?  What are the "heterogeneous dynamics"?
> > >
> > > Since f_S is a simple product, I suggest discussing it in the main paper, not appendix.

---

> > > > ### Author Response · Authors · 2023-11-22
> > > >
> > > > We sincerely appreciate your thoughtful response and thorough feedback. Here, we provide responses to address your concerns
> > > > ## Performance of ConR-only:
> > > > Your concern is completely valid. The key reason lies in the design of RankSim, which is specifically tailored based on an order assumption. This assumption aligns well with label relationships in tasks like age estimation, where RankSim excels.
> > > > However, in more complex regression tasks such as depth estimation, the scenario changes. These tasks do not inherently possess the order relationships that RankSim is designed to leverage. Consequently, RankSim's effectiveness is limited in these contexts due to its specialized focus on order-based relationships, which are less prevalent or entirely absent in complex regression scenarios.\
> > > > Therefore, while "ConR-only" maintains its efficacy across a broader range of tasks, RankSim's specialized design confines its optimal performance to tasks with clear order relationships, like age estimation, but not to more intricate regression challenges.
> > > > ## Representations of ConR vs RankSim
> > > > Your observation regarding the representation of ConR versus RankSim in Figures 4d and 13a is indeed accurate. In these figures, ConR exhibits a larger spread for similar age or colour categories compared to RankSim. This difference can be attributed to the fundamental design of RankSim, which imposes order relationships. Due to this design aspect, RankSim is expected to demonstrate more linear representations, aligning with the linearity of these order relationships.
> > > > However, it's important to note that despite its design for order relationships, RankSim still shows considerable occurrences of feature space collapses. This indicates that while RankSim is effective in maintaining order relationships, it is not immune to certain limitations in its representation capabilities. These collapses in the feature space can be seen as deviations from the expected patterns in the learned representations, highlighting the complexity of the task at hand. As shown in Table 1 and Table 2, despite imposing the order relationships, RankSim does not always outperform ConR in the age estimation task.
> > > > ##  "Feature statistics" and "heterogeneous dynamics"
> > > > In the context of label relationships, 'heterogeneous dynamics' refers to the variability in patterns and features across different label ranges. For instance, in age estimation tasks, ageing patterns are not uniform across all ages, meaning that the features used to estimate age vary significantly [1]. This variability necessitates careful consideration when setting similarity thresholds to avoid incorrectly grouping samples with distinct ageing patterns as similar.\
> > > > Regarding 'feature statistics,' this term generally refers to the distribution of information of features within label ranges. In methods like Feature Distribution Smoothing (FDS) [2], 'feature statistics' might include the mean and covariance of features, which are shared locally to enhance model performance. Essentially, these statistics provide a quantitative description of feature behaviour within specific label ranges.\
> > > > The selection of the similarity threshold, denoted as (ω), plays a crucial role in this context. A lower threshold (or higher 1/ω) means enforcing a less strict criterion for considering features as similar, which can impact how 'positive' and 'negative' pairs are identified in the learning process. This selection directly influences the learned feature space and a proper selection ensures that it accurately reflects the heterogeneous dynamics and feature statistics pertinent to the task at hand.\
> > > > Clarification: In the section titled 'Similarity Threshold Analysis' in our paper, there is a statement: 'An intuitive explanation could be the **higher** thresholds will impose sharing feature statistics to an extent...'. \
> > > > It should be noted that here **higher** refers to 1/ω. This detail was missed in the original submission, leading to potential ambiguity. Including this clarification in the paper would ensure a more precise understanding of our analysis and findings.
> > > > ## Including $f\_{\mathcal{S}}$ in the main paper
> > > > Thank you for your valuable suggestion regarding the discussion of $f\_{\mathcal{S}}$ in the main paper. We appreciate your insight and agree that incorporating this discussion into the main body of the paper could enhance the reader's understanding of our methodology.
> > > > The initial decision to include the details of $f\_{\mathcal{S}}$ in the appendix was primarily driven by space constraints.
> > > > In light of your feedback, we will revisit the paper's structure and explore ways to integrate the discussion of $f\_{\mathcal{S}}$ into the main text without compromising the paper's overall clarity and conciseness.
> > > > ### References
> > > > [1] Li, Wanhua, et al. "Bridgenet: A continuity-aware probabilistic network for age estimation." CVPR 2019.\
> > > > [2] Yang, Yuzhe, et al. "Delving into deep imbalanced regression." ICML 2021.

---

> > > > > ### Comment · Reviewer_Yj3F · 2023-11-22
> > > > > **comments on authors response**
> > > > >
> > > > > What do you consider as "feature space collapses" and "expected patterns in the learned representations" in the age dataset?
> > > > >
> > > > > If there are "heterogeneous dynamics" in the feature space as described, is a constant similarity threshold omega appropriate for all ages?  If so, what is the main reason?

---

> > > > > > ### Author Response · Authors · 2023-11-23
> > > > > >
> > > > > > Thank you very much for your attention to the details of the ConR's design.
> > > > > >
> > > > > > ## What do you consider as "feature space collapses" and "expected patterns in the learned representations" in the age dataset?
> > > > > > "Feature collapses" refer to situations where samples have similar features but dissimilar labels.
> > > > > > By "expected patterns in the learned representations" we mean that although the relative variance in RankSim is better than ConR, there are still many samples with similar representations but dissimilar labels.
> > > > > >
> > > > > > ## If there are "heterogeneous dynamics" in the feature space as described, is a constant similarity threshold omega appropriate for all ages? If so, what is the main reason?
> > > > > > We acknowledge your valid concern and appreciate your insightful question regarding the use of a constant similarity threshold, in the context of 'heterogeneous dynamics' in the feature space.
> > > > > > Indeed, the presence of heterogeneous dynamics introduces a layer of complexity in determining an appropriate threshold that can universally be applied across different label ranges.\
> > > > > > In our current approach, we opted for a constant similarity threshold, primarily to maintain a balance between model simplicity and its effectiveness. The rationale behind setting this threshold as high as possible for modelling locality was to ensure that the model captures the most relevant and proximate feature relationships, which is crucial for the accuracy of our predictions.
> > > > > > However, we acknowledge that this approach might not fully encapsulate the nuanced variations across different age ranges. As such, a promising direction for future research could be to develop a methodology for learning these similarity thresholds adaptively. This would allow the model to dynamically adjust the threshold in response to the varying characteristics of the data, potentially leading to more refined and accurate modelling of the heterogeneous feature space.
> > > > > >
> > > > > > Once again, we extend our deepest thanks for your thorough review and invaluable comments.

---

### Official Review · Reviewer_M28F · 2023-10-28

**Soundness:** 3 good
**Presentation:** 3 good
**Contribution:** 3 good
**Rating:** 8
**Confidence:** 4

**Summary:**

The paper addresses the problem of imbalanced problem in real-world data. To tackle the imbalance in continuous label spaces, the authors introduce a contrastive regularization technique named ConR, which is based on infoNCE. This technique simulates both global and local similarities between labels in the feature space, preventing the features of minority samples from being overshadowed by those of majority samples. ConR primarily focuses on discrepancies between the label space and the feature space and penalizes these differences. Indeed, this is also an augmentation method rather than re-weighted method in DIR.

However, this work contains several weakness, which is discussed below.

**Strengths:**

1. ConR presents a novel auggmentation approach to handle the imbalanced problem in continuous label spaces, whose problem is important.

2. The methods of regularizing process of **ConR** by pulling together positive pairs and relatively repelling negative pairs seems solid.

3. The results seem to indicate that the proposed method can be seamlessly integrated with other models and exhibits improvements over existing baselines.

**Weaknesses:**

1. While the author claims that **ConR** reduces prediction error, are there any theoretical insights or guarantees supporting the idea that **ConR** can achieve a lower generalization bound? Relying solely on empirical results might not suffice to attest to the superiority of the proposed methods.

2. I am curious about complexity. When performing augmentation on large-scale datasets, sampling might increase the complexity. This leads to a prevalent question: Why not leverage reweighting methods which can attain comparable (or potentially superior) results without the significant memory and time overhead associated with the augmentation step? [1,2]

3. While AgeDB-DIR, IMDB-WIKI-DIR, and NYUD2-DIR are structured in DIR [3], and MPIIGaze-DIR is a creation of the authors, a comprehensive description of each dataset should be included in the Appendix. Moreover, given the variety of metrics in NYUD2-DIR, the rationale behind selecting only two needs clarification.

4. The experiments did not surpass all baselines. For instance, in Table 2, the combination of **LDS + FDS + RankSim** posts the best results in terms of GM in few-shot case.

5. For larger datasets like IMDB and NYUD2, **ConR** doesn't consistently outperform other models. However, it seems to excel with smaller datasets such as AgeDB and MPIIGaze.

6. Examining Table 1, the reported results are as follows:

| model | few |
| :---------: | :------: |
|FDS + RankSim|  9.68 |
|FDS + RankSim + **ConR**| 9.43 |
|LDS + FDS + RankSim| 9.92 |
|LDS + FDS + RankSim + **ConR**| 9.21 |
|||

Further analysis is warranted. For instance, why does **LDS + FDS+RankSim** underperform compared to **FDS + RankSim**? Yet, with the addition of **ConR**, it achieves superior results. This observation hints that **ConR** might enhance outcomes when paired with **LDS + FDS + RankSim**. However, this theory doesn't hold when assessed on IMDB-WIKI-DIR. A more detailed analysis of the experimental results is recommended.

[1] Wang et al., Variational Imbalanced Regression: Fair Uncertainty Quantification via Probabilistic Smoothing. NeurIPS 2023

[2] Ren et al., Balanced MSE for Imbalanced Visual Regression, CVPR 2022

[3] Yang et al., Delving into Deep Imbalanced Regression. ICML 2021

**Questions:**

See Above. Overall, this work is novel, and interesting. The problem they try to solve is imporant, and results seems good. Nonetheless, I strongly recommend the authors to delve deeper by:

(1) **theoretical insights or guarantees** supporting the efficacy of the proposed method, and

(2) **thorough empirical analysis of the experimental outcomes**.

---

> ### Author Response · Authors · 2023-11-22
> **Response to Reviewer M28F - part 1**
>
> Thank you for your thorough review and constructive comments. Below, we provide responses to your valuable feedback:
> ## Theoretical Insights
> We appreciate your emphasis on the importance of theoretical analysis. In response to your comment, we have added Section B in the Appendix of the revised manuscript. In the added section, we derive a upper bound on the probability of mislabelling:
>
> $\frac{1}{4N^2}\sum_{j, x_j \in A}^{2N} \sum_{q=0}^{K^-_j} \log \mathcal{S}\_{j,q} \ p(\hat Y\_j | x\_q) \le \mathcal{L}\_{ConR} + \epsilon, \quad \epsilon \overset{N\rightarrow \infty}{\rightarrow} 0$,\
> where $A$ is the set of selected anchors and $x_q$ is a negative sample. $p(\hat Y_j|x_q)$ is the likelihood of sample $x_q$ with an incorrect prediction $y_q \in \hat Y_j = (\hat y\_j - \omega, \hat y\_j + \omega)$. $\hat y\_j$ is the prediction of the $j^{th}$ anchor that is mistakenly similar to the prediction of its negative pairs $x\_q$. We refer to $p(\hat Y_j|x_q)$ as the probability of collapse for $x_q$. The left-hand side presents this probability for all the negative samples.
> The left-hand side is the probability of all collapses during the training and regarding the equation above, Convergence of $\mathcal{L}\_{ConR}$ tightens the upper bound for it. Minimizing the left-hand side can be explained with two non-disjoint scenarios: either the number of anchors or the degree of collapses is reduced. In addition, each collapse probability is weighted with $\mathcal{S}\_{j,q}$, leading to penalizing the incorrect predictions with regard to their severity. In other words, ConR penalizes the bias probabilities with a re-weighting scheme, where the weights are defined based on the agreement of the predictions and labels. This inequality illustrates that optimizing ConR will decrease the probability of mislabelling in proportion to their mislabeling degree, with a specific emphasis on minority samples. Integrating ConR into a regression task penalizes biases, resulting in a more balanced feature space.
>
> ## Leveraging reweighting methods for DIR
> Thank you for highlighting the significance of the complexity analysis. As it is a valid concern, we have addressed it by providing an evaluation of the time consumption of ConR in comparison to the baselines (Please refer to Section 4.1 of the paper, titled "Time Consumption Analysis”). Our analysis show that ConR is considerably more efficient compared to FDS which explicitly aligns the feature space. Moreover, time consumption of ConR is comparable to other baselines while providing considerable performance gain.
>
> Regarding re-weighting, the empirical observations in [1] and highlighted in VIR [2], demonstrate that using empirical label distribution does not accurately reflect the real label density in regression tasks, unlike classification tasks. Thus, traditional re-weighting techniques in regression tasks face limitations. Consequently, LDS [1] and the concurrent work, VIR propose to estimate effective label distribution. Despite their valuable success, two main issues arise, specifically for complicated label spaces. These approaches rely on binning the label space to share local statistics. However, while effective in capturing local label relationships, they disregard global correspondences.
> Furthermore, in complex and high-dimensional label spaces, achieving effective binning and statistics sharing demands in-depth domain knowledge. Even with this knowledge, the task becomes intricate and may not easily extend to different label spaces.
> For instance, in age estimation, where the label space is one-dimensional, methods like LDS and VIR can manage the task. However, when examining the code-base of [1] for depth estimation on the NYUD2-DIR benchmark, we observe that they meticulously define buckets and carefully consider various factors to make LDS feasible in that particular context.
> Additionally, VIR is evaluated on datasets with one-dimensional label space, and the challenges and effectiveness of their method for high-dimensional label spaces are not discussed in their study. In contrast, ConR smoothly extends to high-dimensional label spaces.
> ## Comprehensive description of datasets
> Thank you for emphasizing the importance of a detailed explanation of the datasets. For comprehensive details on the AgeDB-DIR, IMDB-WIKI-DIR, NYUD2-DIR, and the creation process of MPIIGaze-DIR, we have added information to section A.2, and dataset creation details in the Appendix of the updated manuscript.
>
> ### References
> [1] Yang, Yuzhe, et al. "Delving into deep imbalanced regression." ICML 2021.\
> [2] Wang et al., Variational Imbalanced Regression: Fair Uncertainty Quantification via Probabilistic Smoothing. NeurIPS 2023.

---

> > ### Author Response · Authors · 2023-11-22
> > **Response to Reviewer M28F - part 2**
> >
> > ## Evaluation metrics for NYUD2-DIR
> > Thank you for highlighting a valid point about the selection of metrics for NYUD2-DIR. Similar to the baselines, we aimed to avoid overwhelming the tables with an extensive list of metrics and ensures clarity and readability of the results, while still providing a comprehensive evaluation. We can add these metrics to the appendix, if desired.
> > ## Performance analysis of combination of DIR methods
> > We appreciate your attention to the details in the experimental results. We agree that cases, such as the ones the reviewer highlighted, should be thoroughly investigated, and explained.
> > To explain such cases we should consider the nature of the benchmark used for evaluation and the specific design aspects and assumption in the methods. \
> > When examining the results for the age estimation task, it's crucial to consider that RankSim is designed based on the order assumption, aligning with the relationships in the age estimation task. Thus, for certain regions in the label space, imposing the order relationship can yield the better outcomes. However, this method may not be universally applicable to all label spaces.
> > In analyzing cases like the one the reviewer highlighted, where FDS + RankSim outperforms LDS + FDS + RankSim, a thorough examination of the design aspects of the methods is crucial.\
> > FDS and LDS are explicit approaches that locally share statistics in the label space and feature space, respectively. The locality is predefined with the selection of hyperparameters for the proposed kernel smoothing in LDS and FDS. However, assuming the underlying dynamics and localities are the same across all labels would not be accurate. As mentioned in [1], the facial aging process is non-stationary, implying important facts: aging patterns vary in different age ranges (e.g., bone growth in childhood and skin wrinkles in adulthood). This argument emphasizes that at adjacent ages, aging patterns are very similar, aligning with the locality assumption in LDS and FDS. However, an explicit definition of the ranges is not feasible. For example, aging patterns are more non-stationary at early ages, while for seniors, the aging patterns are mostly similar for a broader age range. Thus, we anticipate a smaller similarity range for very young people and larger similarity ranges for seniors. Consequently, using a fixed kernel size in LDS and FDS for all age ranges may face some limitations. Therefore, when LDS is added to the combination of FDS and RankSim, it is possible that its priors do not align with those of RankSim and FDS in some specific label ranges.
> > ## Better Performance on smaller datasets
> > We acknowledge your valid observation. The larger datasets are more challenging and the improvements of DIR methods on these datasets tend to be marginal. Worth mentioning as outlined in our paper, ConR's performance gain is coupled with its efficiency and general-purpose nature, contribute to its significance in the broader context of regression tasks compared to the baselines.
> > ## Comparison of performance improvements across the benchmarks
> > In addressing the notable observation raised by the reviewer, wherein LDS + FDS + RankSim + ConR outperforms in AgeDB-DIR but not in IMDB-WIKI-DIR, we acknowledge that datasets, even within the same task, may exhibit variations that impact the performance of designed methods. Each method aims to address specific challenges; however, subtle differences in dataset characteristics may introduce minor inconsistencies. We appreciate the insight provided by the reviewer and we believe that further investigation of such observations are indeed valuable to understand and address these subtle differences.
> >
> > ### References
> > [1] Li, Wanhua, et al. "Bridgenet: A continuity-aware probabilistic network for age estimation." CVPR 2019.

---

> ### Comment · Reviewer_M28F · 2023-11-22
> **comments on authors response**
>
> Thanks for the authors' comprehensive responses and updates to the manuscript. I would like to suggest the following:
>
> 1. The theoretical insights presented at the end of the appendix are surprising and significant. I recommend incorporating the main theory or conclusion from this insight into the main body of the paper. There is a concern regarding the integration of this material and its impact on the paper's structure and presentation.
>
> 2. The authors offer a detailed comparison between ConR and both [DIR and VIR]. This should be included in the related works section of the main paper, as it provides a clear distinction between these methodologies.

---

> > ### Author Response · Authors · 2023-11-23
> >
> > We are deeply grateful for your thorough review and the insightful suggestions you have provided. Your feedback has been instrumental in enhancing the quality and clarity of our manuscript.\
> > In line with your recommendations, we have integrated the significant theoretical insights previously located in the appendix into the main body of the paper. These insights can now be found in Section 3.3, under 'Method', in the revised manuscript.\
> > Furthermore, we have included a detailed comparison between the ConR approach and the DIR/VIR methodologies in the 'Related Works' section. This addition not only enriches the context of our study but also provides a clearer distinction between these methodologies, aiding readers in understanding the unique contributions of our work.
> >
> > We sincerely thank you again for your thorough and constructive feedback, which has been pivotal in refining our paper. With the incorporation of the theoretical insights into Section 3.3 and the detailed comparison in the 'Related Works' section, we hope to have completely addressed your comments. We look forward to any additional guidance you may offer as we continue to enhance the quality of our work.

---

> ### Comment · Reviewer_M28F · 2023-11-23
> **comments on authors response**
>
> Thank you to the authors for their thorough responses and revisions to the manuscript. In light of these changes, I have increased my scores for **Soundness**, **Presentation**, and **Contribution** as my concerns have been satisfactorily addressed.
>
> However, I would advise the authors to:
>
> **1.** Ensure that concerns raised by other reviewers are also fully addressed. It is important to engage with and resolve these points to strengthen the paper further.
>
> **2.** Regarding the theoretical insights in the appendix, I found no errors or doubts upon review. Nonetheless, I strongly encourage the authors to re-verify the accuracy and completeness of these sections to ensure the robustness of the paper.

---

> > ### Author Response · Authors · 2023-11-23
> >
> > We are extremely grateful for your response and the improved scores. Your acknowledgement of our efforts to address your concerns is highly encouraging.\
> > We assure you that we diligently reviewed and addressed the concerns raised by other reviewers. Each point is being carefully considered and integrated into our revisions to enhance the quality and robustness of our work. We will definitely keep engaging comprehensively with all feedback to ensure the paper's overall strength and integrity. \
> > We also appreciate your advice regarding the theoretical insights in the appendix. While we are confident in our current analysis, your suggestion underscores the importance of thorough verification. We will revisit this section in detail to re-confirm their accuracy and completeness to ensure the robustness and reliability of our paper.
> >
> > Thank you once again for your constructive feedback and guidance that contributed significantly to the quality of our paper.

---

### Official Review · Reviewer_ECg6 · 2023-10-29

**Soundness:** 3 good
**Presentation:** 3 good
**Contribution:** 3 good
**Rating:** 6
**Confidence:** 4

**Summary:**

This paper addresses the imbalanced regression problems where the label space is continuous. The proposed contrastive regularizer is to model global and local label similarities in feature space and prevent the features of minority samples from being collapsed into their majority neighbours. The proposed ConR consolidates essential considerations into a generic, easy-to-integrate, and efficient method that effectively addresses deep imbalanced regression. The empirical study shows that ConR significantly boosts the performance of SoTA methods on four large-scale deep imbalanced regression benchmarks.

**Strengths:**

1. This paper is well-written and easy to understand.
2. The novelty of this paper is good. To my knowledge, applying contrastive learning to imbalance regression is novel.
3. The authors provide a new dataset with 2-dimentional label space by using MPIIGaze, which could be useful to the imbalance regression community.

**Weaknesses:**

1. The proposed objective in Eq. (4) is relative hasty, without the reason of introducing the ordinary regression loss. Also, the definition of the introduced loss should be clearly provided.
2. No deviation measure in the experimental results.
3. No experimental result for the ConR-only case. All the results for the proposed ConR are with respect to the combination with existing methods as a regularized. Due to supervised information is already used in the loss of ConR, its preformation should be provided as a baseline.
4. Some references are missing, e.g., Page 16.

**Questions:**

1. In the selection of anchor, if an example without any negative example, it will not be chosen as an anchor. What is the main reason for this selection? Do you consider the contrastive learning method without negative pairs, such as BYOL and SimSiam?
2. The empirical label distribution is needed to determine the pushing weight for the negative pair. In addition, the authors suggest using the inverse frequency to compute the pushing weight, so that the minority samples will obtain harder force to be repelled from the anchor. How can we determine the continuous weight with the discrete frequency? Do we need any kernel density estimation technique?
3. Please explain the specific reason for combining the proposed loss with the ordinary regression loss in Eq. (4) with more details and evidence.
4. I notice some failure case in Table 1, Table 2, and Table 3, when adding the ConR loss as the regularization term. It is strange that you can down-weight of $\beta$ to avoid this situation. I understand that the main goal is to improve the performance on *Few* case, however, there are still some cases that the performance on *Few* case drops. Also, there are some cases that the performance on *All* case drops. Hence, please provide the explanation for this phenomenon and what is the specific metric for a good imbalanced regression in your paper. I think the *Few* case should be much more important.
5. Why the results on the *Few* case of Balanced MSE in Figure 3 is inconsistent to that in Table 3?

---

> ### Author Response · Authors · 2023-11-22
> **Response to Reviewer ECg6 - part 1**
>
> Your comprehensive review and insightful comments are greatly appreciated. Here we address your comments as follows:
> ## Combining ConR with the ordinary regression loss
> Thank you very much for your attention to the details of the objective function. We acknowledge that we should provide more information. First, Eq. 4 emphasis that ConR is orthogonal to deep regression methods and can be used in any regression framework with any regression objective function.  Secondly, to clarify the objective, let's delve into the optimization process in the DIR task. When modeling inter-label relationships in the feature space, two crucial factors come into play: correct similarities and correct directions [1]. Taking the age estimation task as an example, consider a scenario with a 20-year-old, a 15-year-old, and a 25-year-old person. Both the 15-year-old and 25-year-old individuals have the same label similarity with the 20-year-old (i.e., a 5-year age difference); however, one is younger, and one is older. ConR imposes proper similarities, with a focus on minority samples, while ordinary regression boosts the direction signals and result in the best performance for ConR. Furthermore, RankSim [2] is also a regularizer, assuming order relationships and optimizing their proposed loss function alongside the regression loss function.
> ## Deviation measure in the experimental results
> We appreciate your valid concern. To maintain clarity in the dense tables of DIR evaluations, we opted not to include deviations directly, aiming to prevent the tables from becoming overwhelming. Instead, we have visualized the standard deviations using error bars in Figure 3 and Figure 11. We can add a detailed table of standard deviations to the supplementary material if desired.
> ## Performance of ConR-only as a baseline
> Thank you very much for highlighting this. We value your concern and added ConR-only to the tables.
> ## Missing reference
> Thank you for highlighting the missing reference. This is fixed in the updated manuscript.
> ## Main reason for anchor selection
> We appreciate your inquiry regarding the rationale behind the proposed anchor selection.
> Yes, ConR does not select the input sample without any negative pair as an anchor. But the selected anchors have both negative and positive pairs dissimilar to BYOL and SimSiam.
> The motivation of this design is multifaceted. First, as discussed in the manuscript, for categorical contrastive learning, the anchors are predefined, and it serves as a prior knowledge in addressing imbalanced distribution. However, in contrastive learning for regression, defining anchors is not feasible, especially for complicated and high-dimensional label spaces.
> Secondly, treating each training sample as an anchor overlooks the imbalanced distribution, resulting in a bias towards majority labels. Empirical observations, as detailed in [3], highlight a phenomenon in deep imbalanced regression where minority samples tend to collapse towards their neighboring majority samples. This occurs even when their labels are dissimilar; for instance, in the age estimation task, features representing the age range of 0 to 6 years may exhibit similarities with those of age 30. This situation leads to confusions within certain areas of the feature space. Thus, we propose to use the correspondences between label similarities and prediction similarities to identify the confusion regions during the training. These areas of confusion are selected as anchors and push the samples that mistakenly have similar features to these anchors. The reason is that due to the skewed distribution, if we select all samples as anchors and repel all negative pairs, the learning will be biased toward majority samples. Therefore, ConR uses the anchor selection as a treatment to the biases in deep imbalance regression task without any prior knowledge about the anchors or any explicit assumption about the inter-label relationships.
> In summary, for each training sample $x_i$ with label $y_i$ and prediction $\hat y_i$, other training samples with a label different to $y_i$ and a prediction close to $\hat y_i$ are considered as collapses due to imbalanced distribution and serve as negative samples for $x_i$. If $x_i$ has no negative pair, it means there are no confusion around $(x_i,y_i)$ to be addressed; thus $x_i$ is not considered as an anchor and does not contribute to ConR. Otherwise, $x_i$ is an anchor and ConR pushes away samples collapsed to $x_i$ relative to label similarities to address these confusions.
>
> ### References
> [1] Li, Wanhua, et al. "Bridgenet: A continuity-aware probabilistic network for age estimation." CVPR 2019.\
> [2] Gong, et al. "RankSim: Ranking similarity regularization for deep imbalanced regression." ICML 2022.\
> [3] Yang, Yuzhe, et al. "Delving into deep imbalanced regression." ICML 2021.

---

> > ### Author Response · Authors · 2023-11-22
> > **Response to Reviewer ECg6 - part 2**
> >
> > ## Continuous weight with the discrete frequency
> > Thank you for your attention to this important aspect. Any weighting approach for the regression task like the inverve frequency and square-root weighting(SQINV) [1] can be used. For continuous label space these techniques bin the label space and compute the per-bin frequencies. For a fair comparison to the baseline we used SQINV for all the experiments. A kernel density estimation is not a necessary option for ConR. LDS [1] uses kernel density estimation to improve the weightings for regression tasks and can improve the sample weights when combined with weighting techniques like SQINV.
> > ## Evaluation metrics
> > We appreciate your attention to the details in the experimental results. To explain such failure cases we should consider the nature of the benchmark used for evaluation and the specific design aspects and assumption in the methods. When examining the results for the age estimation task, it's crucial to consider that RankSim [2] is designed based on the order assumption, aligning with the relationships in the age estimation task. Thus, for certain regions in the label space, imposing the order relationship can yield the better outcomes. However, this method may not be universally applicable to all label spaces. Worth mentioning as outlined in our paper, ConR's performance gain is coupled with its efficiency and general-purpose nature, contribute to its significance in the broader context of regression tasks compared to the baselines.
> > About the evaluation process, few-shot region and after that median region are the most important metrics. However, well-performant DIR method should not harm the performance on all- and many-shots. All the approaches for handling DIR exhibit failure cases. For instance, Balanced MSE demonstrates improved performance for few- and median-shots but experiences degradation in performance for many-shot cases. Each method tries to address specific aspects of the DIR problem, yet the complexities inherent in the task make these failure cases unavoidable. Consequently, each method contributes to some extent and there remains potential for future advancements in tackling these challenges.
> > ## Inconsistency between Figure 3 and Table 3
> > Thank you for your detailed review, and we acknowledge the mentioned inconsistencies. There was an error in reporting the results for Balanced MSE+ConR in the table 3, and in Figure 3. We have addressed these issues in the updated manuscript.
> >
> > ### References
> > [1] Yang, Yuzhe, et al. "Delving into deep imbalanced regression." ICML 2021.\
> > [2] Gong, et al. "RankSim: Ranking similarity regularization for deep imbalanced regression." ICML 2022.

---

### Official Review · Reviewer_viAu · 2023-10-31

**Soundness:** 2 fair
**Presentation:** 3 good
**Contribution:** 2 fair
**Rating:** 6
**Confidence:** 4

**Summary:**

The paper introduces a contrastive learning approach to address the issue of imbalanced regression. This method is orthogonal to existing solutions and has demonstrated promising experimental results.

**Strengths:**

1. Data imbalance and the fairness of machine learning algorithms are practical issues that warrant significant attention.
2. The method is reasonably designed and has shown good experimental results.

**Weaknesses:**

1. While the paper presents a contrastive learning paradigm adapted for regression, it does not appear to directly address the issue of data imbalance. It would enhance the paper if the authors could clarify how the method specifically tackles this challenge or consider adapting the technique to more explicitly focus on imbalanced datasets.
2. Comparisons should be made with other contrastive regression learning methods (e.g., [a]).
3. The manuscript could be strengthened by providing some theoretical analysis and insights to support the empirical findings.
4. It would be beneficial if the authors could provide the pseudocode for the algorithm to facilitate the readers' comprehension of the algorithm's details.

[a] [Rank-N-Contrast: Learning Continuous Representations for Regression](https://arxiv.org/pdf/2210.01189.pdf)

**Questions:**

Given the distinct advantages of data augmentation under the contrastive learning framework, are the baseline methods utilizing their typical data augmentation strategies, or those consistent with contrastive learning? If it's the former, a comparison showcasing results after aligning the augmentation techniques would be valuable.

---

> ### Author Response · Authors · 2023-11-22
> **Response to Reviewer viAu - part 1**
>
> Thank you sincerely for your comprehensive review and invaluable insights. In the following we provide the response to your comments:
> ## Effectiveness of ConR for Deep Imbalanced Regression
> We appreciate your attention to the contribution of ConR to the imbalanced learning. Here we provide clarifications as follows:
> Contrastive learning techniques have shown promising potentials in learning semantically rich representations by effectively disentangling essential variations from irrelevant ones. Emphasizing contrastive optimization in favor of minority samples to encourages balanced feature space is a common practice in imbalanced classification[1]. These approaches employ predefined anchors and statistical priors that are not extendible to the regression tasks.
> By incorporation continuity into the contrastive learning, the variations in the feature space will be modeled according to the continuity in the label space. Consequently, each sample is accurately mapped to the feature manifold in agreement with its label region on the label manifold, making it easier for the model to traverse the regions where data might be scarce due to imbalanced distribution.
> In addition, ConR emphasizes on modeling continuous relationships for minority samples via dynamically selecting the region of confusion (The anchor selection and negative pair selection in ConR) in the feature space to enforce variations in accordance with the label similarities (The relative pushing in ConR).
> Moreover, in section B of appendix (Added to the updated manuscript), as theoretical analysis shows, by optimizing ConR, the probability of incorrect labeling specifically for minority samples is reduced and ConR helps regularizing the regression process.
> ## Comparison with Rank-N-Contrast
> Thank you for your valuable suggestion regarding the comparison with Rank-N-Contrast [2].  We added Rank-N-Contrast as a concurrent work in our updated manuscript (Please refer to our latest manuscript). Their code became available in October, which is after our ICLR 2024 submission deadline, making a direct comparison infeasible. We acknowledge the importance of comprehensive comparisons; however, providing comparative results takes longer than the short timeframe of the Discussion period. Nonetheless, we plan to include 'Rank-N-Contrast' as a concurrent baseline and provide the necessary comparison results.
> Additionally, 'Rank-N-Contrast' is designed based on the order assumption in the label space, similar to RankSim[3]. As mentioned in our paper, this assumption is not universally valid for all label spaces. For instance in depth estimation task, 'Rank-N-Contrast' and RankSim cannot be used.  However our proposed method, ConR, does not make any assumptions about inter-label relationships.
> ## Theoretical Insights
> We appreciate your emphasis on the importance of theoretical analysis. In response to your comment, we have added Section B in the Appendix of the revised manuscript. In the added section, we derive a upper bound on the probability of mislabelling:
>
> $\frac{1}{4N^2}\sum_{j, x_j \in A}^{2N} \sum_{q=0}^{K^-_j} \log \mathcal{S}\_{j,q} \ p(\hat Y\_j | x\_q) \le \mathcal{L}\_{ConR} + \epsilon, \quad \epsilon \overset{N\rightarrow \infty}{\rightarrow} 0$,\
> where $A$ is the set of selected anchors and $x_q$ is a negative sample. $p(\hat Y_j|x_q)$ is the likelihood of sample $x_q$ with an incorrect prediction $y_q \in \hat Y_j = (\hat y\_j - \omega, \hat y\_j + \omega)$. $\hat y\_j$ is the prediction of the $j^{th}$ anchor that is mistakenly similar to the prediction of its negative pairs $x\_q$. We refer to $p(\hat Y_j|x_q)$ as the probability of collapse for $x_q$. The left-hand side presents this probability for all the negative samples.
> The left-hand side is the probability of all collapses during the training and regarding the equation above, Convergence of $\mathcal{L}\_{ConR}$ tightens the upper bound for it. Minimizing the left-hand side can be explained with two non-disjoint scenarios: either the number of anchors or the degree of collapses is reduced. In addition, each collapse probability is weighted with $\mathcal{S}\_{j,q}$, leading to penalizing the incorrect predictions with regard to their severity. In other words, ConR penalizes the bias probabilities with a re-weighting scheme, where the weights are defined based on the agreement of the predictions and labels. This inequality illustrates that optimizing ConR will decrease the probability of mislabelling in proportion to their mislabeling degree, with a specific emphasis on minority samples. Integrating ConR into a regression task penalizes biases, resulting in a more balanced feature space.
>  ### References
> [1] Cui, Jiequan, et al. "Parametric contrastive learning.", ICCV 2021.\
> [2] Zha, Kaiwen, et al. "Rank-N-Contrast: Learning Continuous Representations for Regression." Neurips 2023.\
> [3] Gong, et al. "RankSim: Ranking similarity regularization for deep imbalanced regression." ICML 2022.

---

> > ### Author Response · Authors · 2023-11-22
> > **Response to Reviewer viAu - part 2**
> >
> > ## Pseudocode for the algorithm
> > We acknowledge the importance of including the pseudocode. In response to your comment we added the pseudocode of the proposed algorithm in section C of the appendix in the revised version.
> > ## Augmentation Analysis
> > Thank you for highlighting the importance of selecting appropriate augmentations for contrastive learning. In our evaluation, we examined the impact of data augmentations on ConR, particularly when applied to the AgeDB-DIR benchmark for LDS+ConR. During this investigation, we observed that ConR exhibits overall robustness to augmentations, yielding only marginal performance gains with regression-specific augmentations.
> > To probe the underlying cause, we looked through the differences between the “contrastive learning-specific” augmentations and “Age estimation-specific” augmentations. The most commonly used augmentations in contrastive learning [1] are "RandomHorizontalFlip", "RandomGrayscale, "ColorJitter", "RandomResizedCrop" and "GaussianBlur". The Augmentations for age estimation in the baselines [2] for AgeDB-DIR dataset consist of "RandomHorizontalFlip" and "RandomCrop".
> > Among all “contrastive learning-specific” augmentations, we assumed that "GaussianBlur" is not a feasible choice for the age estimation task. The reason is that blurring reduces the distinctive facial features crucial for age estimation, such as wrinkles.
> > Thus, to empirically validate our assumption we excluded GaussianBlur from "contrastive learning-specific" augmentations and used the rest to evaluate ConR. ConR with these augmentations performed comparably to 'age estimation-specific' augmentations and exhibited superior performance compared to when "GaussianBlur" was included.
> > Table 1 and Table 2 show the results of our experiments.
> > Next, for evaluating the distinct advantages of data augmentations, we chose the set of "contrastive learning-specific" augmentations without "GaussianBlur". Then, we compared the performance of the VANILLA model (baseline regression model with no DIR technique) with VANILLA + ConR, using these augmentations for both VANILLA and VANILLA+ConR. The results are shown in Table 1 and Table 2 and confirms that performance gain by ConR is considerably higher than just modifying the augmentations. Results are reported as the average of 5 random runs.
> >
> > Table 1: MAE results
> > | Method             |          Augmentation set         |           All        |  Many | Median | Few |
> > |---|--------------------------|---|---|---|---|
> > | LDS + ConR| "Age estimation-specific" |7.16 |	6.61|	7.97	|9.62|
> > | LDS + ConR| "contrastive learning-specific" |7.22|	6.72|	8.01|	9.71|
> > | LDS + ConR| "Contrastive learning-specific without GaussianBlur" |7.18|	6.68	|7.92|	9.66|
> > | VANILLA| "Age estimation-specific" |7.77	|6.62	|9.55|	13.67|
> > | VANILLA +ConR| "Age estimation-specific" |7.20| 6.50 |8.04| 9.73|
> > | VANILLA| "Contrastive learning-specific without GaussianBlur" |7.82|	6.56	|9.73	|13.87|
> > | VANILLA +ConR| "Contrastive learning-specific without GaussianBlur" |7.27|6.54|7.99|9.81|
> >
> > Table 2: GM results
> > | Method             |          Augmentation set         |           All        |  Many | Median | Few |
> > |---|--------------------------|---|---|---|---|
> > | LDS + ConR| "Age estimation-specific" |4.51	|4.21	|4.92|	5.87|
> > | LDS + ConR| "contrastive learning-specific" |4.59	|4.25	|5.00|	6.10|
> > | LDS + ConR| "Contrastive learning-specific without GaussianBlur" |4.53	|4.19	|4.93|	5.86|
> > | VANILLA| "Age estimation-specific" |5.05	|4.23	|7.01	|10.75|
> > | VANILLA +ConR| "Age estimation-specific" |4.59| 3.94 |4.83 | 6.39|
> > | VANILLA| "Contrastive learning-specific without GaussianBlur" |5.12	|4.01	|6.99	|10.69|
> > | VANILLA +ConR| "Contrastive learning-specific without GaussianBlur" |4.62|3.88|4.79|6.41|
> >
> >
> >
> >
> > We didn’t include these experiments in the submitted manuscript, as we believed that these might not significantly contribute to the overall discussion. However, if the reviewer suggests that more comprehensive evaluation on augmentations for all benchmarks is insightful, we will further include them in the Appendix.
> > Considering these observations and recognizing the potential influence of augmentation choices on regression performance, we have chosen to follow the baselines and maintain task-specific augmentations in our experiments for a fair comparison to the baselines.
> >
> > ### References
> > [1] He, Kaiming, et al. "Momentum contrast for unsupervised visual representation learning." CVPR 2020.\
> > [2] Yang, Yuzhe, et al. "Delving into deep imbalanced regression." ICML 2021.

---

> > > ### Comment · Reviewer_viAu · 2023-11-23
> > >
> > > Thanks for the response. After reading the comments and other reviewer's comments, my major concerns have been addressed. I will increase the score accordingly.
> > >
> > > Best,
> > >
> > > The reviewer viAu

---

> > > > ### Author Response · Authors · 2023-11-23
> > > >
> > > > Thank you for your acknowledgement and for increasing the score. We're grateful to hear that your concerns have been addressed. Your feedback has been invaluable in improving our paper.

---

### Comment · Area_Chair_maiT · 2023-11-22
**Author-Reviewer Discussion ends soon**

Dear Authors! The discussion phase ends soon. This is your last chance to send your rebuttal.

Dear Reviewers! If the Authors will send their rebuttal, please try to respond to it (at least by acknowledging that you have read it).

Thank you!

Best, AC for Paper #2927

---

> ### Comment · Reviewer_Yj3F · 2023-11-30
>
> Three reviewers rated "marginally above the threshold" and one rated "accept".  I won't be championing the paper.

---

### Meta-Review · Area_Chair_maiT · 2023-12-11

**Metareview:**

The paper concerns the problem of label imbalance in regression problems. The authors introduce a contrastive regularizer that models global and local label similarities preventing the features of less-represented labels from being collapsed into their frequent-label neighbors. The method has obtained promising results in the experimental studies.

**Justification For Why Not Higher Score:**

The paper received three 6s and one 8. The topic, although interesting, is rather of limited interest.

**Justification For Why Not Lower Score:**

All reviewers agree that the paper is above the acceptance threshold.

---

### Decision · Program_Chairs · 2024-01-16

Accept (poster)